# Double Bubble, Toil and Trouble: Enhancing Certified Robustness through Transitivity

**Andrew C. Cullen**[1*]    **Paul Montague**[2]    **Shijie Liu**[1]
**Sarah M. Erfani**[1]    **Benjamin I.P. Rubinstein**[1]
[1]School of Computing and Information Systems, University of Melbourne, Parkville, Australia
[2]Defence Science and Technology Group, Adelaide, Australia
`andrew.cullen@unimelb.edu.au`

## Abstract

In response to subtle adversarial examples flipping classifications of neural network models, recent research has promoted certified robustness as a solution. There, invariance of predictions to all norm-bounded attacks is achieved through randomised smoothing of network inputs. Today's state-of-the-art certifications make optimal use of the class output scores at the input instance under test: no better radius of certification (under the $L_2$ norm) is possible given only these score. However, it is an open question as to whether such *lower bounds* can be improved using local information around the instance under test. In this work, we demonstrate how today's "optimal" certificates can be improved by exploiting both the transitivity of certifications, and the geometry of the input space, giving rise to what we term *Geometrically-Informed Certified Robustness*. By considering the smallest distance to points on the boundary of a set of certifications this approach improves certifications for more than $80\%$ of Tiny-Imagenet instances, yielding an on average $5\%$ increase in the associated certification. When incorporating training time processes that enhance the certified radius, our technique shows even more promising results, with a uniform $4$ percentage point increase in the achieved certified radius.

## 1   Introduction

Learned models, including neural networks, are well known to be susceptible having the output changed by crafted perturbations to an input, that preserve the inputs semantic properties [2]. Neural networks not only misclassify these perturbations—known as *adversarial examples*—but they also assign high confidence to these incorrect predictions. These behaviours have been observed across a wide range of models and datasets, and appear to be a product of piecewise-linear interactions [13].

Crafting these adversarial examples typically involves gradient-based optimisation to construct *small* perturbations. These attacks have been applied to both black- and white-box models [31], and can be used to target class changes, to attack all classes [10], or even introduce backdoors into model behaviour [5]. To mitigate the influence of these attacks, defences have typically been designed to minimise the effect of a specific attack (or attacks). Such defences are known as *best response* strategies in a Stackelberg security game where the defender leads the attacker. Best response defences inherently favour the attacker, as deployed mitigations can be defeated by identifying undefended attack frameworks. Moreover, the defender typically has to incorporate the defence at training time, and as such cannot response reactively to newly developed attacks.

To circumvent these limitations, certified guarantees of adversarial robustness can be constructed to identify class-constant regions around an input instance, that guarantee that all instances within a norm-bounded distance (typically $L_2$) are not adversarial examples. Certifications based on randomised smoothing of classifiers around an input point are in a sense optimal [8]: based only on the prediction

36th Conference on Neural Information Processing Systems (NeurIPS 2022).

class scores at the input point, no better radius is in general possible. Despite this, such certifications fail to use readily available—yet still local—information: the certifiability of points nearby to the input of interest. The key insight of this work is that these neighbourhood points may generate certified radius large enough to completely enclose that of a sample point, improving the radius of certification. This process can be extended to use the intersection of the regions of certification of multiple points, and the nature of the input domain itself to generate larger certifications. This leads to our main contribution—*Geometrically-Informed Certified Robustness*—that produces certifications exceeding those of the hitherto best-case guaranteed approach of Cohen *et al.* (2019) [8].

## 2 Background and literature review

**Bounding mechanisms** Conservative bounds upon the impact of norm-bounded perturbations can be constructed by way of either Interval Bound Propagation (IBP) which propagates interval bounds through the model; or Convex Relaxation, which utilise linear relaxation to construct bounding output polytopes over input bounded perturbations [34, 26, 41, 45, 46, 37, 28], in a manner that generally provides tighter bounds than IBP [25]. In contrast to Randomised Smoothing, bounding mechanisms employ augmented loss functions during training, which promote tight output bounds [42] at the cost of decreased applicability. Moreover they both exhibit a time and memory complexity that makes them infeasible for complex model architectures or high-dimensional data [40, 6, 21].

**Randomised smoothing** Outside of bounding mechanisms, another common framework for developing certifications leverages *randomised smoothing* [20], in which noise is applied to input instances to smooth model predictions, subject to a sampling distribution that is tied to the $L_P$-norm of adversarial perturbations being certified against. In contrast to other robustness mechanisms, this application of the noise is the only architectural change that is required to achieve certification. In the case of $L_2$-norm bounded attacks, Gaussian sampling of the form

$$\mathbf{x}'_i = \mathbf{x} + \mathbf{y}_i \qquad \text{where } \mathbf{y}_i \overset{i.i.d.}{\sim} \mathcal{N}(0, \sigma^2) \ \ \forall i \in \{1, \ldots, N\} \tag{1}$$

is employed for all test-time instances. These $N$ samples are then used to estimate the expected output of the predicted class of $\mathbf{x}$ by way of the Monte-Carlo estimator

$$E_{\mathbf{Y}}[\arg\max f_\theta(\mathbf{x} + \mathbf{Y}) = i] \approx \frac{1}{N} \sum_{j=1}^{N} \mathbb{1}[\arg\max f_\theta(\mathbf{x}_j) = i] \ . \tag{2}$$

While this Monte Carlo estimation of output expectations under randomised smoothing is a test-time process, model sensitivity to random perturbations may be decreased by performing adversarial training on such random perturbations. To mitigate the computational expense of large $N$ sample sizes during each training update, training typically employs single draws from the noise distribution.

**Smoothing-based certifications** Based on randomised smoothing, certified robustness can guarantee classification invariance for *additive* perturbations up to some $L_p$-norm $r$, with recent work also considering rotational and/or translational semantic attacks [23, 7]. $L_p$-norm certifications were first demonstrated by way of differential privacy [20, 11], with more recent approaches employing Rényi divergence [22], and parametrising worst-case behaviours [8, 33]. By considering the worst-case $L_2$-perturbations, Cohen *et al.* (2019) purports that the largest achievable pointwise certification is

$$r = \frac{\sigma}{2} \left( \Phi^{-1} \left( E_0[\mathbf{x}] \right) - \Phi^{-1} \left( E_1[\mathbf{x}] \right) \right) \ . \tag{3}$$

Here $\{E_0, E_1\}$ are the two largest class expectations (as per Equation (2)), $\sigma$ is the noise, and $\Phi^{-1}$ is the inverse normal CDF, or Gaussian quantile function.

## 3 Geometrically-informed certified robustness

While the work contained within this paper can be applied generally, for this work we will focus upon certifications of robustness about $L_2$-norm bounded adversarial perturbations, for which we assume that the difficulty of attacking a model is proportional to the size of the certification, based upon the need to evade both human *and* machine scrutiny [12]. Thus, constructing larger certifications in such a context is inherently valuable.

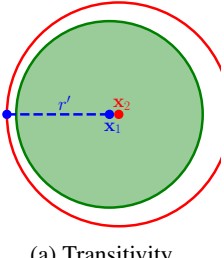 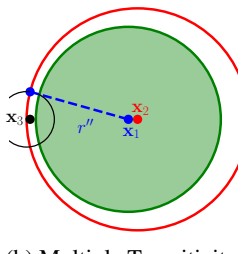 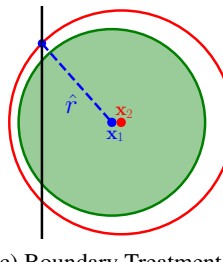

(a) Transitivity      (b) Multiple Transitivity     (c) Boundary Treatment

Figure 1: Transitive certification exemplars. The Green, Red, and Black circles represent hyperspheres of radius $r_i$ (by Equation 3) about points $\mathbf{x}_i \forall i \in \{1, 2, 3\}$. The resulting certifications $r'$, $r''$, and $\hat{r}$ are described within Equations 5, 12, and 16. The Black line represents the domain boundary.

This specific $L_P$ space is of interest due to both its viability as a defence model, and the provable guarantee that Cohen *et al.* produces the largest possible certification for any instance [8]. Over the remainder of this section we will document how it is possible to improve upon this provably best-case guarantee by exploiting several properties of certified robustness.

## 3.1 Exploiting transitivity

While it is provably true that Equation (3) is the largest achievable certification for any point $\mathbf{x}$, it is possible to exploit the behaviour of points in the neighbourhood of $\mathbf{x}$ in order to enhance certifiable radius. To achieve this, consider the case of a second point $\mathbf{x}'$, that exists within the certifiable radius of $\mathbf{x}$. As both points must correspond to the same class, it then follows that the union of their regions of certification can be also be considered as a region of certification, leading to Definition 3.1.

**Definition 3.1 (Overlap Properties of Certification).** A radius of certification $r_i$ about $\mathbf{x}_i$ can be calculated by evaluating Equation 3 at $\mathbf{x}_i$. This certification guarantees that no point $\mathbf{x} : \|\mathbf{x} - \mathbf{x}_i\|_P \leq r_i$ can induce a chance in the predicted class. That this shape is a $d$-dimensional hypersphere for input data $\mathbf{x} \in \mathcal{R}^d$ allows us to introduce the notational shorthand

$$B_P(\mathbf{x}_i, r_i) = \{\mathbf{x} \in \mathcal{R}^d | \|\mathbf{x} - \mathbf{x}_i\|_P \leq r_i\} \qquad S_i = \{\mathbf{x} \in \mathcal{R}^d | \|\mathbf{x} - \mathbf{x}_i\|_P = r_i\} \qquad (4)$$

to represent the region covered by the hypersphere and its surface. It follows from this definition that that if $B_P(\mathbf{x}_1, r_1) \cap B_P(\mathbf{x}_2, r_2) \neq \emptyset$, which ensures that the class predictions at $\mathbf{x}_1$ and $\mathbf{x}_2$ match, then the region of certification about $\mathbf{x}_1$ can be expressed as $B_P(\mathbf{x}_1, r_1) \cup B_P(\mathbf{x}_2, r_2)$.

However typically we are concerned not with the size of the region of classification invariance, but rather the distance to the nearest adversarial example. If it is possible to find some $\mathbf{x}'$ such that its region of certification completely encircles that of the certification at $\mathbf{x}$, the following definition demonstrates that the certification radius about $\mathbf{x}$ can be increased.

**Lemma 3.2 (Set Unions Certified Radius).** *If $\mathbf{x}_1$ and $\mathbf{x}_2$ have the same class associated with them and $B_P(\mathbf{x}_1, r_1) \subset B_P(\mathbf{x}_2, r_2)$, then the nearest possible adversarial example—and thus, the certifiable radius—exists at a distance $r' \geq r$ from $\mathbf{x}_1$, where*

$$r' = r_2 - \|\mathbf{x}_2 - \mathbf{x}_1\|_P \quad , \qquad (5)$$

*Proof.* The closest point on the surface of $B_P(\mathbf{x}_2, r_2)$ to $\mathbf{x}_1$ must exist on the vector between $\mathbf{x}_1$ and $\mathbf{x}_2$. Thus $r' = \min\left(r_2 \pm \|\mathbf{x}_2 - \mathbf{x}_1\|_P\right)$, which takes the form of Equation (5). □

As such, we can recast the task of constructing a certification from being a strictly analytic function to the nonlinear optimisation problem in terms of a second ball with certified radius $r_2$ centred at $\mathbf{x}_2$

$$r' = \max_{\mathbf{x}_2 \in [0,1]^d} r_2 - \|\mathbf{x}_2 - \mathbf{x}_1\|_P \qquad (6)$$

with Figure 1a providing a two-dimensional exemplar. Crucially, the above formalism does not require obtaining a global optima, as any $r' > r_1$ yields an improved certification at $\mathbf{x}_1$.

## 3.2 Multiple transitivity

To further enhance our ability to certify, let us consider the set of points and their associated certifications $\{\mathbf{x}_1, r_1, \mathbf{x}_2, r_2, \ldots \mathbf{x}_n, r_n\}$. If the union of $\hat{B}_P = \cup_{i \in \{1,\ldots,n\}} B_P(\mathbf{x}_i, r_i)$ is simply-connected, then the certification across this set can be expressed as $r^{(n)'} = \min_{\mathbf{x} \in \partial \hat{B}_P} \|\mathbf{x} - \mathbf{x}_1\|_P$, where $\partial \hat{B}_P$ is the boundary of $\hat{B}_P$. This can be further simplified by imposing that $\mathbf{x}_j \in B_P(\mathbf{x}_2, r_2) \forall j > 2$ and that $B_P(\mathbf{x}_j, r_j) \not\subset B_P(\mathbf{x}_k, r_k) \; \forall (j > 2, k > 2)$ to ensure that hyperspheres exist near the boundary of $\mathcal{S}_2$ and yielding a certification of

$$r^{(n)'} = \min_{\mathbf{x} \in \mathbf{S}_n} \|\mathbf{x} - \mathbf{x}_1\| \text{ where } \mathbf{S}_n = \mathcal{S}_2 \cap \left( \cup_{i=3}^{n+1} S_i \right) \text{ for } n \geq 2 \ . \tag{7}$$

Here $\mathbf{S}_n$ is a $(d-1)$-dimensional manifold embedded in $\mathcal{R}^d$.

**Lemma 3.3 (Optimal positioning of $\mathbf{x}_3$ in the case of $n = 3$ ).** *Consider the addition of a new hypersphere at some point $\mathbf{x}_3$ with associated radius $r_3$, which has an associated boundary $\mathcal{S}_3$. If it is true that*

$$B_P(\mathbf{x}_3, r_3) \not\subset B_P(\mathbf{x}_2, r_2) \text{ and } B_P(\mathbf{x}_3, r_3) \cap B_P(\mathbf{x}_2, r_2) \neq \emptyset \tag{8}$$

$$B_P(\mathbf{x}_3, r_3) \not\subset B_P(\tilde{\mathbf{x}}, \tilde{r}) \; \forall \{\tilde{\mathbf{x}} \in [0,1]^d | \tilde{\mathbf{x}} \neq \mathbf{x}_3\} \text{ with } \tilde{r} = \frac{\sigma}{2} \left( \Phi^{-1}(E_0[\tilde{\mathbf{x}}]) - \Phi^{-1}(E_1[\tilde{\mathbf{x}}]) \right) \ , \tag{9}$$

*then the largest possible certification $r''$ by Equation 7 is achieved at*

$$\mathbf{x}_3(s) = \mathbf{x}_1 + sr' \frac{\mathbf{x}_1 - \mathbf{x}_2}{\|\mathbf{x}_1 - \mathbf{x}_2\|_2} \qquad \textbf{\textit{for some}} \ s \in [0,1] \ . \tag{10}$$

*Proof.* The closest point to $\mathbf{x}_1$ upon $\mathcal{S}_2$ is located at

$$\check{\mathbf{x}} = \mathbf{x}_1 + r' \frac{\mathbf{x}_1 - \mathbf{x}_2}{\|\mathbf{x}_1 - \mathbf{x}_2\|_2} \ , \tag{11}$$

where $r'$ is defined by Equation 6. Thus any improved radius of certification $r'' > r'$ is only achievable if $B_P(\mathbf{x}_3, r_3)$ satisfies $\check{\mathbf{x}} \in B_P(\mathbf{x}_3, r_3)$ and Equation 8. Then by symmetry, $r''$ is the maximally achievable radius of certification if Equation 9 hold and if $\mathbf{x}_3$ is defined by Equation 10. $\qquad \square$

While finding some $\mathbf{x}_3$ satisfying Equations 8 and 10 is trivial, proving Equation 9 would require an exhaustive search of the input space $[0,1]^d$. However, even in the absence of such a search, Equation 10 still provides the framework for a simple search for $\mathbf{x}_3$, which follows Figure 1b.

**Lemma 3.4 (Certification from two eccentric hyperspheres).** *If $\mathbf{x}_3$ is defined by Equation 10 in a fashion that satisfies Equation 8 then an updated certification can be achieved in terms of some $\mathbf{x}_3(s)$ defined by Equation 10 by way of*

$$r'' = \max_{s \in [0,1]} \sqrt{\frac{d_2(r_3^2 - d_3^2) + d_3(r_2^2 - d_2^2)}{d_2 + d_3}} \text{ where}$$

$$d_2 = \|\mathbf{x}_2 - \mathbf{x}_1\| \qquad d_3 = \|\mathbf{x}_3(s) - \mathbf{x}_1\| \qquad r_3 = \frac{\sigma}{2} \left( \Phi^{-1} \left( E_0[\mathbf{x}_3(s)] \right) - \Phi^{-1} \left( E_1[\mathbf{x}_3(s)] \right) \right) \ . \tag{12}$$

*If Equation 9 holds, then this is the largest achievable certification for $n = 3$.*

*Proof.* By symmetry we can define the arbitrary rotational mapping $f : \mathcal{R}^d \to \mathcal{R}^d$ from $\mathbf{x}_i \mapsto \mathbf{y}_i$ by way of $\mathbf{y}_i = f(\mathbf{x}_i - \mathbf{x}_1)$, subject to the condition

$$y_{i,j} \neq 0 \text{ for } i \in \{1, 2\} \text{ and } \forall j \neq k, \text{ for some } k \text{ such that } j, k \in \{1, 2, \ldots, d\} \tag{13}$$

then the intersection of the hyperspheres centred about $\mathbf{x}_2$ and $\mathbf{x}_3$ occurs at

$$\begin{aligned} r_3^2 - r_2^2 &= \|\mathbf{y} - \mathbf{y}_3\|^2 - \|\mathbf{y} - \mathbf{y}_2\|^2 \\ &= 2y_{2,k}(d_2 + d_3) + d_3^2 - d_2^2 \\ 2y_{2,k} &= \frac{r_3^2 - r_2^2 + d_2^2 - d_3^2}{d_2 + d_3} \ . \end{aligned} \tag{14}$$

This is a consequence of our mapping $y = f(\mathbf{x} - \mathbf{x}_1)$ preserving distances under rotation, giving that $y_{2,k} = \|\mathbf{x}_2 - \mathbf{x}_1\| = d_2$, and with the equivalent also holding for $y_{3,k}$.

As a consequence of our choice of coordinate system, it follows that $r'' = \|\mathbf{y}\|$ and

$$
\begin{aligned}
(r'')^2 &= \|\mathbf{y} - \mathbf{y}_2\|^2 + 2y_{2,k}d_2 - d_2^2 \\
&= r_2^2 - d_2^2 + \frac{d_2 r_3^2 - d_2 r_2^2 + d_2 d_2^2 - d_2 d_3^2}{d_2 + d_3}
\end{aligned}
\tag{15}
$$

which is an equivalence to Equation (12). $\qquad\square$

While Equation 7 holds for any $n \geq 2$, the certification radius beyond $n = 3$ cannot be enhanced by adding any one single additional hypersphere without contradicting Lemma 3.4. This is a result of $\mathbf{S}_3$ being a $(d-1)$-dimensional manifold in $\mathcal{R}^d$, the entirety of which must be enclosed to improve the certification. An example of this can be seen with the two equidistant intersections between $\mathcal{S}_2$ (in Red) and $\mathcal{S}_3$ (in Black) in Figure 1b. While multiple spheres could be constructed to completely enclose $\mathbf{S}_3$, the number required grows exponentially with $d$ due to the sphere packing kissing number problem [9]. This growth in complexity makes adding additional spheres beyond $n = 3$ infeasible. Further details of this are contained within Appendix A.2.

### 3.3 Boundary treatments

Without loss of performance or accuracy, we can freely scale the inputs of neural networks such that $\mathbf{x}_1 \in [0,1]^d$. However in the majority of cases a subset of $B_P(\mathbf{x}_1, L_1)$ will exist outside $[0,1]^d$. While this observation is trivially true, it has no influence on the radius of certification achieved by Equation 3 due to the symmetry of $B_P(\mathbf{x}_1, r)$. However, the asymmetric nature of $B_P(\mathbf{x}_2, r_2)$ about $\mathbf{x}_1$ guarantees that if $B_P(\mathbf{x}_2, r_2)$ exceeds $[0,1]^d$, then the closest point to $\mathbf{x}_1$ within the feasible domain must have an associated distance $\hat{r} > r'$, as is demonstrated within Figure 1c. This allows us to make the following observation about improving the feasible radius of certification.

**Lemma 3.5** (**Boundary Certifications by way of Eccentric Circles**). *The eccentricity of $B_P(\mathbf{x}_2, L_2)$ as a bounding region about $\mathbf{x}_1$, and the potential for a subset of $B_P(\mathbf{x}_2, L_2)$ to exist outside the feasible space for instances allows us to construct an updated region of certification where*

$$
\hat{r} = \max_{k \in \{1,\ldots,d\}} \mathbb{1}\left[ r_2^2 - (x_{2,k} - z_k)^2 \geq 0 \right] \times
$$

$$
\sqrt{(z_k - x_{1,k})^2 + \left( \sqrt{r_2^2 - (z_k - x_{2,k})^2} - \sqrt{\|\mathbf{x}_2 - \mathbf{x}_1\|^2 - (x_{2,k} - x_{1,k})^2} \right)^2}
\tag{16}
$$

*where $z_k = \begin{cases} 0 & \text{if } x_{1,k} < 0.5 \\ 1 & \text{if } x_{1,k} > 0.5 \end{cases}$,*

*where $\mathbb{1}$ is an indicator function acting upon its operator.*

*Proof.* In contrast to the prior proof, for this problem we retain the coordinate system of the input space. To support this, we introduce the notation that $\mathbf{x}_n = \{x_{n,1}, x_{n,2}, \ldots, x_{n,d}\}$. If we let $x_k \in \{0,1\}$, then the intersection between $B_P(\mathbf{x}_2, L_2)$ and the bounding surface in dimension $k$ creates a bounding hypersphere of the form

$$
\sum_{i=1, i \neq k}^{d} (x_i - x_{2,i})^2 = r_2^2 - (x_k - x_{2,k})^2 \ .
\tag{17}
$$

which yields an effective radius $\widetilde{r}_2 = \sqrt{r_2^2 - (z_k - x_{2,k})^2}$.

By denoting the projection of $\mathbf{x}_1$ and $\mathbf{x}_2$ onto the bounding hyperplane in the $k$-th dimension as $\widetilde{\mathbf{x}_1}$ and $\widetilde{\mathbf{x}_2}$, then the distance from $\mathbf{x}_1$ to Equation (17) must take the form

$$
\begin{aligned}
\hat{r_k} &= \sqrt{\|\widetilde{\mathbf{x}_1} - \mathbf{x}_1\|^2 + (\widetilde{r}_2 - \|\widetilde{\mathbf{x}_2} - \widetilde{\mathbf{x}_1}\|)^2} \\
&= \sqrt{(z_k - x_{1,k})^2 + \left( \sqrt{r_2^2 - (z_k - x_{2,k})^2} - \sqrt{\|\mathbf{x}_2 - \mathbf{x}_1\|^2 - (x_{2,k} - x_{1,k})^2} \right)^2} \ .
\end{aligned}
\tag{18}
$$

By imposing that $z_k = 1$ when the $k$-th component of $\mathbf{x}_1$ is greater than 0.5, and 0 otherwise, it follows that $\max_{k \in \{0,1,\ldots,d\}} z_k$ must be an improved radius of certification. $\qquad\square$

---

**Algorithm 1** Single Bubble Loop.

---

 1: **Input:** data $\mathbf{x}$, samples $N$, iterations $M$, true-label $i$
 2: **for** 1 **to** $M$ **do**
 3:    $\hat{E}_0$, $\hat{E}_1$, $L_2$, $j$ = Algorithm 2 $\left(\mathbf{x}' + \mathcal{N}\left(0, \sigma^2 \mathcal{I}_N\right), N, \sigma\right)$
 4:    **if** $j = i$ **then**
 5:       $r' = L_2 - \|\mathbf{x}' - \mathbf{x}\|$
 6:       **if** $r' > r'_o$ **then**
 7:          $r'_o = r'$
 8:       **end if**
 9:       $\mathbf{x}' = \mathbf{x}' \pm \gamma \frac{\nabla_{\mathbf{x}'} r'}{\|\nabla_{\mathbf{x}'} r'\|}$ $\{\gamma$ calculated by Barzilai-Borwein [1]. Positive branch is selected if
           $j = i$, otherwise the negative branch brings $\mathbf{x}'$ towards the region in which $j = i.\}$
10:    **end if**
11: **end for**

---

### 3.4 Algorithms

To demonstrate how the above certification approaches can be applied in practice, Algorithm 1 demonstrates the application of Equation (5) through a simple, gradient based solver. Such a solver is highly applicable for solving such problems, due to the inherent smoothing nature of randomised smoothing being applicable both to the function space and its derivatives. To implement the multiple transitivity based approach of Section 3.2, Algorithm 1 can trivially be adapted to evaluate derivatives with respect to Equation (12). The boundary treatment of Section 3.3 does not require any additional calculations, but instead is simply the result of applying Equation (16) to the output of Algorithm 1.

## 4 Extracting gradient information from non-differentiable functions

Implementing the aforementioned process requires the ability to evaluate the gradient of the class expectations. This is problematic, as each class expectation is described in terms of a finite sum of non-differentiable indicator functions, as is seen in Equation (2). Within this work we have implemented two mechanisms to circumvent these conditions. The first substitutes the $\arg\max$ operation with a Gumbel-Softmax [15]. In doing so, the class expectations are rendered differentiable.

The second approach involves recasting the Monte-Carlo estimators as integrals of the form

$$E[\arg\max f_\theta(\mathbf{x}) = k] = \int \mathbb{1}[\arg\max f_\theta(\mathbf{y}) = k]\omega(\mathbf{y}, \mathbf{x})d\mathbf{y} \ , \tag{19}$$

where $\omega(\mathbf{y}, \mathbf{x})$ is the multivariate-Normal probability distribution centred around $\mathbf{x}$. This formalism, and the symmetry of the underlying space allows for the construction of undifferentiable gradients by

$$\begin{aligned}
\nabla_{\mathbf{x}} E[\arg\max f_\theta(\mathbf{x}) = k] &= \nabla_{\mathbf{x}} \int \mathbb{1}[\arg\max f_\theta(\mathbf{y}) = k]\omega(\mathbf{y}, \mathbf{x})d\mathbf{y} \\
&\approx \int \mathbb{1}[\arg\max f_\theta(\mathbf{y}) = k]\nabla_{\mathbf{x}}\omega(\mathbf{y}, \mathbf{x})d\mathbf{y} \\
&= \frac{1}{\sigma^2} \int \mathbb{1}[\arg\max f_\theta(\mathbf{y}) = k](\mathbf{y} - \mathbf{x})\omega(\mathbf{y}, \mathbf{x})d\mathbf{y} \\
&\approx \frac{1}{N\sigma^2} \sum_{i=1}^{N} \mathbb{1}[\arg\max f_\theta(\mathbf{x}_i) = k](\mathbf{x}_i - \mathbf{x}) \\
&\text{where } \mathbf{x}_i = \mathbf{x} + \mathcal{N}(0, \sigma^2) \ .
\end{aligned} \tag{20}$$

While this derivation is novel, the resultant gradient operator reflects those seen in prior works [33]. It is important to note that such a sampling process is inherently noisy, and it has previously suggested that the underlying uncertainty scales with the input dimensionality [27].

The relative performance of these two approaches—respectively labelled 'Approximate' and 'Full' for the above approach and the Gumbel-Softmax approaches—will be tested in the following section. For the case of the double transitivity of Section 3.2 our experiments suggest that uncertainty in the

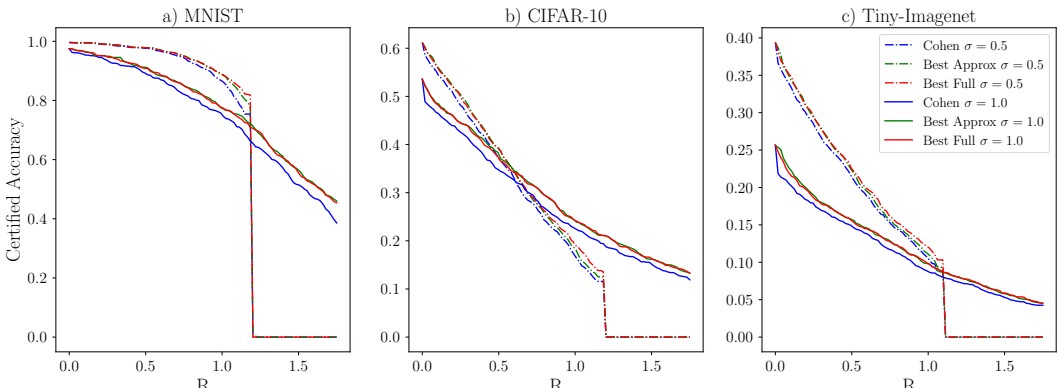

Figure 2: The Certified Accuracy: the proportion of samples correctly predicted and with a certified radius greater than $R$. Blue represents Cohen, while Green and Red respectively represent the best sub-variant utilising either the Approximate or Full derivative approaches. Dashed lines for $\sigma = 0.5$, solid lines for $\sigma = 1.0$.

analytic derivatives produces deleterious results. As such derivatives for the multiple transitivity approach are exclusively considered through autograd for both the Full and Approximate methods.

## 5 Experiments

**Configuration**   To evaluate the performance of our proposed certification improvements, we considered the certified radius produced for MNIST [19], CIFAR-10 [18], and Tiny-Imagenet [16], the latter of these is a 200-class variant of Imagenet [43] which downsamples images to $3 \times 60 \times 60$. All datasets were modelled using the Resnet18 architecture in PyTorch [32], with Tiny-Imagenet also utilising 2D adaptive average pooling.

For both MNIST and CIFAR-10, our experimentation utilised a single NVIDIA P100 GPU core with 12 GB of GPU RAM, with expectations estimated over 1500 samples. Training employed Cross Entropy loss with a batch size of 128 over 50 epochs. Each epoch involved every training example was perturbed with a single perturbation drawn from $\mathcal{N}(0, \sigma^2)$, which was added prior to normalisation. Parameter optimisation was performed with Adam [17], with the learning rate set as 0.001. Tiny-Imagenet training and evaluation utilised 3 P100 GPU's and utilised 1000 samples. Training occurred using SGD over 80 epochs, with a starting learning rate of 0.1, decreasing by a factor of 10 after 30 and 60 epochs, and momentum set to 0.9.

The full code to implement our experiments can be found at https://github.com/andrew-cullen/DoubleBubble.

**Certified accuracy**   To explore the performance advantage provided by our technique, we begin by considering the performance of Cohen *et al.* against the best of our approaches using both the approximate and full derivatives, as seen in Figure 2. While there are only minor differences between the two derivative approaches, there are clear regions of out performance relative to Cohen across all tested datasets. The proportion of this increase appears to be tied to the semantic complexity of the underlying dataset, with decreases in predictive accuracy (visible at $R = 0$) appearing to elicit greater percentage changes in the achieved certified radius, as evidenced by the results for Tiny-Imagenet.

Semantic complexity drives a decrease in the overall confidence of models, inducing a decrease in the separation between the highest class expectations $E_0$ and $E_1$. While this process shrinks the achievable certified radius, a higher $E_1$ provides more information to inform our gradient based search process, allowing for the identification of larger certifications. This property would suggest that samples with a larger $R$ would exhibit a decreased difference between our techniques and that of Cohen *et al.* would decrease, as smaller values of $E_1$ provide less search information. However, it appears that singularities in the derivatives of $\Phi^{-1}[E]$ as $E \to \{0, 1\}$ counteract the decreased information provided by the second highest class, leading to the contradictory performance best observed in the MNIST experiments of Figure 2 at $\sigma = 0.5$.

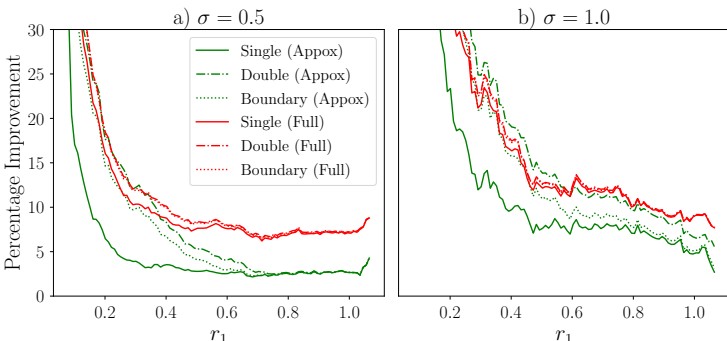

Figure 3: Percentage improvement in the Certified Radius of Tiny-Imagenet instances relative to Cohen *et al.*. for varying $r$. This measure presents the median improvement over $[r-0.075, r+0.075]$. Equivalent figures for MNIST and CIFAR-10 are found in Appendix A.3

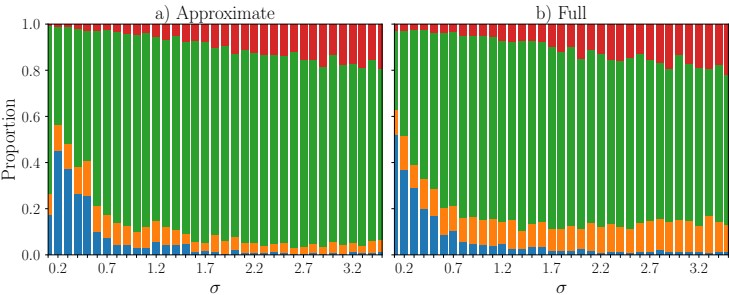

Figure 4: Proportion of correctly predicted instances for which each approach yields the highest certification for Tiny-Imagenet. Red, Green, and Orange represent the boundary treatment, Double transitivity, and Single transitivity, with both ties Cohen *et al.* in Blue. MNIST and CIFAR-10 results can be seen in Appendix A.3

Rather than strictly considering the best performing of the Approximate and Full solvers, we can also delve into the relative performance of the underlying solvers. Notably there is a moderate increase in the average percentage improvement of Figure 3 between the $\sigma = 0.5$ and $\sigma = 1.0$ cases. This would appear to belie our previous statement regarding larger certifications yielding smaller improvements, due to the asymmetry of class information. However, an equivalent certification for $\sigma = 1.0$ has significantly more information about the second class (due to the multiplicative influence of $\sigma$), allowing for greater improvements from our gradient based search. That there is a clear demarcation between the Full and Approximate solver variants reflects the uncertainties introduced by Equation (20). That the approximate technique is still applicable verifies the utility of our approach even when the final layer is a strict $\arg\max$ function, rather than a Gumbel-Softmax.

The trends in performance across $\sigma$ are further explored in Figures 4 and 6, the latter of which demonstrates that the median performance improvement increases quasi-linearly with $\sigma$. This is driven by both an increase in the performance of the certifications themselves, and in the number of instances able to be certified in an improved fashion. This later property stems from the smoothing influence of $\sigma$, with larger levels of added noise inducing decreases in the difference between the highest class expectations, improving the ability for our search based mechanisms to identify performance improvements. Here increases in the performance of the boundary treatment are correlated with larger radii of certification, due to the multiplicative influence of $\sigma$ upon Equation (3).

**Numerical performance** The numerical optimisation process at the core of our certification process inherently induces increases in the computational cost of finding these improved certifications, as shown in Table 1. While analytically approximating the derivatives for the first eccentric hypersphere yields a lower certified accuracy, the fact that the corresponding computational cost decreases by a factor of more than 4 emphasises the value of this approach. Interestingly, while the Approximate

Table 1: Average wall clock time (in seconds) for each computational technique, for a single sample evaluated over 1000 draws under noise.

| Dataset | | Approx. | | | Full | | |
|---------|-------|--------|----------|--------|--------|----------|--------|
| | Cohen | Single | Boundary | Double | Single | Boundary | Double |
| M | 0.08 | 0.66 | 0.66 | 2.94 | 2.17 | 2.17 | 3.97 |
| C-10 | 0.08 | 0.76 | 0.76 | 3.83 | 2.08 | 2.08 | 4.62 |
| T-I | 0.12 | 1.13 | 1.13 | 5.61 | 2.87 | 2.86 | 7.95 |

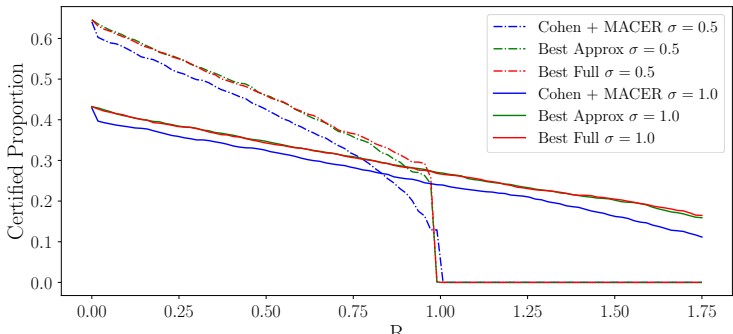

Figure 5: Certified Accuracy comparing the Cohen certification (when trained incorporating MACER) for CIFAR-10, as well as the best sub-variant utilising either the Approximate or Full derivative approaches. Dashed lines for $\sigma = 0.5$, solid lines for $\sigma = 1.0$.

method does also utilise auto-differentiation for the Double variant, the increase in computational cost from the Single to Double variants is significantly higher than for the Actual approach. This is surprising, as the Approx variant derives smaller values of $\mathbf{r}'$, which should in turn lead to a smaller, easier to navigate search space for $\mathbf{r}''$. Instead it counter-intuitively appears that a smaller $\mathbf{r}'$ induces a search space for $\mathbf{r}''$ which is less convex, and more difficult to converge upon.

**Alternative training routines** Recent work has considered the potential for enhancing certified robustness by modifying the underlying training regime to incentivise maximising the expectation gap between classes [33]. One such approach is MACER [44], which augments the training time behaviour by considering not just the classification loss, but also the $\epsilon$-robustness loss, which reflects proportion of training samples with robustness above a threshold level. Such a training time modification can increase the average certified radius by $10 - 20\%$, however doing so does increase the overall training cost by more than an order of magnitude.

When applying Geometrically-Informed Certified Robustness to models trained with MACER, Figure 5 demonstrates that our modifications yield an even more significant improvement than those observed in Figure 2. Under training with MACER, the best performing of our techniques yielded an approximately 4 percentage point increase in the average certification. From this it is clear that while MACER does enhance the certified radii at a sample point, it also induces enough smoothing in the neighbourhood of the sample point to allow transitivity to yield even more significant improvements than are present without MACER.

However, we must emphasise that while such training time modifications do producer consistently larger certifications, doing so requires significantly more computational resources, both in terms of training time and GPU memory, as compared to the more traditional certification training regime. We also emphasise that training with MACER requires a degree of re-engineering. In contrast the training mechanism used for the remainder of this work only requires the addition of noise to samples prior to being passed through the model, and thus imposes significantly fewer engineering considerations.

**Limitations** While the principles of our Geometrically Informed Certified Robustness are extensible to $L_p$ spaces, our experimental work has so far only considered $L_2$-norm bounded perturbations due to the guarantee of best possible certification in this space provided by Cohen *et al.*. Further

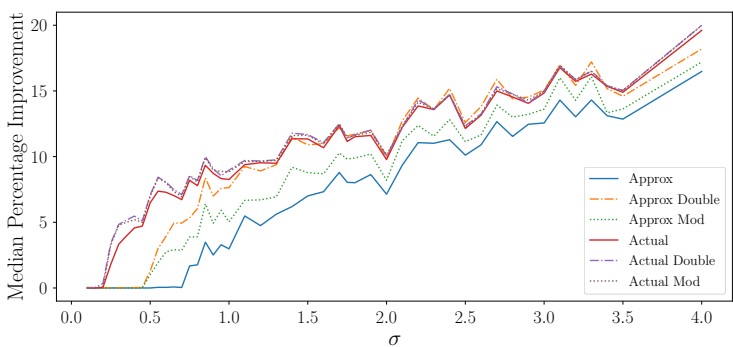

Figure 6: Median percentage improvement in the Certified Radius achieved by each of our approaches relative to Cohen *et al.* for Tiny-Imagenet across the level of additive noise $\sigma$. The median was chosen to provide a fair and representative comparison to Cohen *et al.*, that filters out outliers in the percentage improvement when $r \ll 1$.

experimentation could also consider both these general spaces and a broader range of training methods, which have been shown to be able to tighten the achievable radius of certification [24].

We note that enhanced robustness certification have the potential to counter beneficial applications of adversarial attacks, such as those used to promote stylometric privacy [3]. However, we believe this drawbacks is significantly outweighed by the potential for enhanced confidence about models for which adversarial incentives exist.

Finally, we also emphasise that our approach requires $M$ evaluations of the certified robustness, which each require $N$ Monte-Carlo draws, resulting in time- and memory-complexity of $\mathcal{O}(MN)$ and $\mathcal{O}(NS)$ respectively, where $S$ is the size of the output logit vector. With respect to the memory-complexity, this is shared by any randomised smoothing based approach, and could be improved by implementing batching across the Monte-Carlo process. While the time cost can be problematic in some contexts, we emphasise that this framework is both requires both fewer adaptations to the training loop and significantly less training time relative to bound propagation approaches [21, 36]. We believe these costs can be reduced by performing the optimisation stages with $N' < N$ model draws, and by potentially reusing model draws across the iterative process to approach the $\mathcal{O}(N)$ time-complexity of prior randomised smoothing based certifications. Even at present, we believe that the increased computational cost is not intractable, especially for human-in-the-loop certifications.

## 6   Acknowledgements

This research was undertaken using the LIEF HPC-GPGPU Facility hosted at the University of Melbourne. This Facility was established with the assistance of LIEF Grant LE170100200. This work was also supported in part by the Australian Department of Defence Next Generation Technologies Fund, as part of the CSIRO/Data61 CRP AMLC project. Sarah Erfani is in part supported by Australian Research Council (ARC) Discovery Early Career Researcher Award (DECRA) DE220100680.

## 7   Conclusions

This work has presented mechanisms that can be exploited to improve achievable levels of certified robustness, based upon exploiting underlying geometric properties of robust neural networks. In doing so our Geometrically Informed Certified Robustness approach has been able to generate certifications that exceed prior guarantees by on average more than $5\%$ at $\sigma = 0.5$, with the percentage increase improving quasi-linearly with $\sigma$. Incorporating training time modifications likes MACER yields more promising results, with the best performing of our approaches yielding a $4$ percentage point increase in the certified proportion at a given radius. Being able to improve upon the size of these guarantees inherently increases the cost of constructing adversarial attacks against systems leveraging machine learning, especially in the case where the attacker has no ability to observe the size of the robustness certificate.

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
