**Algorithm 2** Class prediction and certification, as required for Algorithm 1
___
1: **Input:** Perturbed data $\mathbf{x}'$, samples $N$, level of added noise $\sigma$
2: $\mathbf{y} = \mathbf{0}$
3: **for** i = 1:N **do**
4:    $y_j = y_j + 1$ if $GS\left(f_\theta\left(\mathbf{x}' + \mathcal{N}(0, \boldsymbol{\sigma}^2)\right)\right) = j$ {Here $GS$ is the Gumbel-Softmax}
5: **end for**
6: $\mathbf{y} = \frac{1}{N}\mathbf{y}$
7: $z_0, z_1 = \text{topk}(\mathbf{y}, k = 2)$ {topk is used as it is differentiable, $z_0 > z_1$}
8: $\breve{E}_0, \widehat{E}_1 = \text{lowerbound}(\mathbf{y}, z_0), \text{upperbound}(\mathbf{y}, z_)1$ {Calculated by way of Goodman *et al.* [14]}

9: $R = \frac{\sigma}{2}\left(\Phi^{-1}(\breve{E}_0) - \Phi^{-1}(\widehat{E}_1)\right)$
10: **return** $\breve{E}_0, \widehat{E}_1, R, j$
___

# A  Appendix

## A.1  Algorithmic details

Algorithm 2 supports Algorithm 1 by demonstrating how the class prediction and expectations are calculated. Of note are two minor changes from prior implementations of this certification regime. The first is the addition of the Gumbel-Softmax on line 4, although this step is only required for the 'Full' derivative approach. In contrast th 'Approximate' techniques able to circumvent this limitation and can be applied directly to the case where the class election is determined by an $\arg\max$.

The second difference to prior works is the calculation of the lower and upper bounds on $z_0$ and $z_1$ on line 8. Our initial testing revealed that when we employed either Sison-Glaz [38] or Goodman *et al.* [14] to estimate the multivariate class uncertainties, some Tiny-Imagenet samples devoted more than $95\%$ of their computational time of the process to evaluating the confidence intervals, significantly outweighing even the costly process of model sampling. Further investigation revealed that this was occurring when there were a significant number of classes reporting counts of approximately 0, the likelihood for which was higher in Tiny-Imagenet due to the increased class count relative to MNIST and CIFAR-10. To resolve this, we coalesced all classes where $y_j < 5$ into one single meta-class with an associated class-count $y' = \max(5, \sum_{y_j < 5} y_j$, which conforms with the requirements of Goodman *et al.* [14] that all class counts must be greater than 5. Our testing demonstrated that while this process slightly decreased the resulting radius of certification (due to small changes in $z_0$ and $z_1$), the associated decrease in computational time was significant enough to justify this modification.

We also note that all the experiments contained within this work have been conducted against publically releaed datasets with established licenses. MNIST exists under a GNU v3.0 license; CIFAR-10 employs a MIT license; and Imagenet employs a BSD $3-$Clause license.

## A.2  Ramifications of the dimensionality for $n > 3$

To improve the achieved certification in the case $n > 3$, the added set of hyperspheres must fully enclose the $(d-1)$-dimensional manifold that marks the intersection between $\mathcal{S}_2$ and $\mathcal{S}_3$. In two-dimensions—as is used in the examplar Figure 1a—this intersection takes the form of two points. If Lemma's 3.3 and 3.4 are to hold, then encompasing $\mathcal{S}_2 \cap \mathcal{S}_3$ will require two additional certification hyperspheres to be identified.

In the case where $d = 3$, the intersection between these two hyperspheres is the boundary of the circle (equivalent to a $d = 2$ hypersphere) with radius

$$\tilde{r} = \sqrt{r_2^2 - \frac{\|\mathbf{x}_3 - \mathbf{x}_2\|_2 - r_3^2 + r_2^2}{2\|\mathbf{x}_3 - \mathbf{x}_2\|_2}} \quad . \tag{21}$$

Thus any set of spheres $\{\mathcal{S}_2, \mathcal{S}_3, \ldots, \mathcal{S}_n\}$ must uniformly cover all points on the boundary of this surface if we seek to improve the achieved certification.

To provide an indicative example of how the complexity of the region that must be encircled grows with the underlying dimensionality, we now consider some properties of hyperspheres. In higher

dimensions, prior work [39] has demonstrated that the volume contained within a $d$-dimensional hypersphere can be expressed as

$$V_d(\tilde{r}) = \frac{\tilde{r}^d \pi^{d/2}}{\Gamma\left(\frac{d}{2}+1\right)} \quad, \tag{22}$$

with an associated surface area of

$$\begin{aligned} S_d(\tilde{r}) &= \left.\frac{d\left(V_d(r)\right)}{d\left(r\right)}\right|_{r=\tilde{r}} \\ &= \frac{d\tilde{r}^{d-1}\pi^{d/2}}{\Gamma\left(\frac{d}{2}+1\right)} \quad. \end{aligned} \tag{23}$$

Thus if $S_2$ and $S_3$ are $d$-dimensional hyperspheres, then their region of intersection would in turn be a $(d-1)$-dimensional hypersphere, the exterior boundary of which scales with $\tilde{r}^{d-2}$. While $\tilde{r}$ may be less than 1, it should also be true that any additional spheres would likely have an associated radii less than $\tilde{r}$. As such there would appear to be a power-law proportionality with respect to $(d-1)$ between the area covered by the intersection manifold and the size of spheres over which we would seek to enclose said manifold. This underscores the complexity of finding a set of hyperspheres to encircle the boundary of $\mathcal{S}_2 \cap \mathcal{S}_3$.

To give further evidence in the growth of complexity, let us consider a unit-hypersphere in $\mathcal{R}^d$ that represents the intersection of two hyperspheres in $\mathcal{R}^{d+1}$. The task of covering such a hypersphere is similar to that of the sphere packing kissing number $k(d)$ [9], which describes the number of touching-but-not-overlapping unit-hyperspheres that can exist upon the surface of a $d$-dimensional hypersphere. To date, the kissing number has only been solved for the following dimensions outlined in Table 2, however it has been shown to exhibit exponential growth [4].

Table 2: Known Kissing numbers $k(d)$ for $d$-dimensional hyperspheres

| $d$ | 1 | 2 | 3 | 4 | 8 | 24 |
|---|---|---|---|---|---|---|
| $k(d)$ | 2 | 6 | 12 [35] | 24 [29] | 240 [30] | $196,560$ [30] |

Within the context of this work, the kissing number must be considered to be a significant underestimate of the number of boundary spheres that would be required to be found, as we must cover all the space around the central sphere (rather than just maximising the number of hyperspheres without intersection), and it is unlikely that the smallest of the set of encircling hyperspheres has the same radius as the region to be encircled. As such, we can be highly confident that the growth in complexity of the task of enclosing the boundary of intersection beyond the set of hyperspheres $\{\mathcal{S}_1, \mathcal{S}_2, \mathcal{S}_3\}$ is exponential.

We must also note that even if it were possible to perform such a bounding operation, the gains in certified radius would be exceedingly minor. If the region of intersection between $\mathcal{S}_2$ and $\mathcal{S}_3$ was a hypersphere of radius $\check{r}$, then going from the case where $n = 3$ to $n > 3$ would only increase the certified radius from $\sqrt{(r')^2 + (\check{r})^2}$ to $\sqrt{(r')^2 + 9(\check{r})^2}$, which is trivial relative to the increase in computational complexity.

### A.3 Relative performance for MNIST and CIFAR-10

While Figure 2 presents the best performing certified accuracy, it is important to understand the relative performance of the Single Transitivity, Double Transitivity, and Boundary treatments, in a similar fashion to Figures 3 and 4. In the case of MNIST, while the percentage increases exhibited in Figure 7 as $r_1 \to 1$ are broadly similar to their Tiny-Imagenet counterpart for the Approximate solver, the difference between those results and the Full derivative treatment is significantly smaller, especially at $\sigma = 1$. This may, in part, be driven by the 1500 samples employed when using MNIST and CIFAR-10, in contrast to 1000 for Tiny-Imagenet, which should decrease the uncertainty of the gradient estimation steps.

However, the fact that this decreased difference in performance holds for CIFAR-10 at $\sigma = 1.0$ but not $\sigma = 0.5$ suggests that the performance difference between the techniques is also dependent upon the semantic complexity of the prediction task. While CIFAR-10 is a more complex predictive

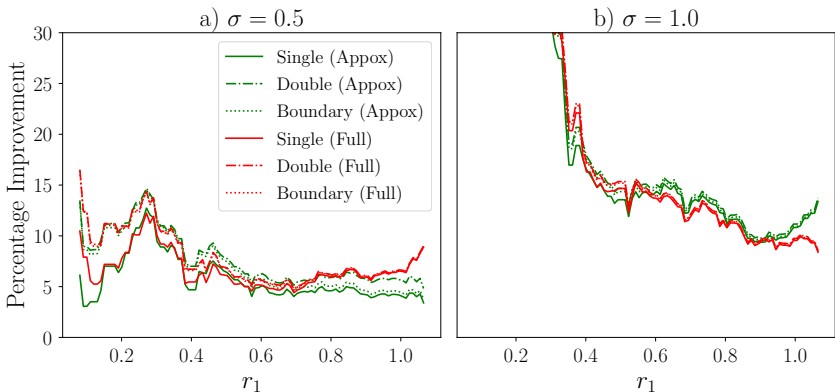

Figure 7: Percentage improvement in the Certified Radius of Tiny-Imagenet instances relative to Cohen *et al.*. for varying $r$. This measure presents the median improvement over $[r-0.075, r+0.075]$. Equivalent figures for Tiny-Imagenet can be seen in Figure 3

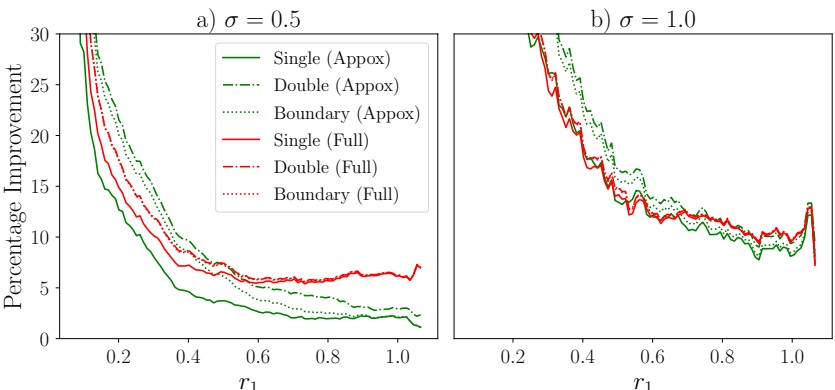

Figure 8: Percentage improvement in the Certified Radius of CIFAR-10 instances relative to Cohen *et al.*. for varying $r$. This measure presents the median improvement over $[r - 0.075, r + 0.075]$. Equivalent figures for Tiny-Imagenet can be seen in Figure 3

environment than MNIST, which should increase the complexity of the gradient based search routine employed within this work, the increased level of noise at $\sigma = 1.0$ has a smoothing influence that decreases the complexity of the search task, and it would appear that this is the primary driver of the relative under performance of the Approximate derivatives in both Tiny-Imagenet and CIFAR-10 when $\sigma = 0.5$.

When considering the median percentage improvement (relative to Cohen *et al.*) of these techniques, MNIST again reveals interesting properties when we consider Figure 11. When compared to CIFAR-10 and Tiny-Imagenet (in Figures 12 and 6) it becomes apparent that the Approximate approach only produces consistently larger certifications in MNIST. Given the increased uncertainty in the derivatives calculated by the Approximate technique, this would suggest that the approximate solver may be improved by considering common improvements to gradient descent methods like momentum or the addition of calibrated noise.

While MNIST may be the simplest of all the prediction tasks, Figure 9 demonstrates that at low $\sigma < 0.5$ the majority of samples cannot be improved upon by any of the certification enhancements developed within this paper. Given that this does not hold for CIFAR-10 nor Tiny-Imagenet (in Figure 10 and 4 respectively) this would suggest that the potential for Cohen *et al.* to be improved upon in low semantic complexity datasets is smaller. That this behaviour is predominantly seen for small $\sigma$ also suggests that our initial step size may be too large in these particular cases.

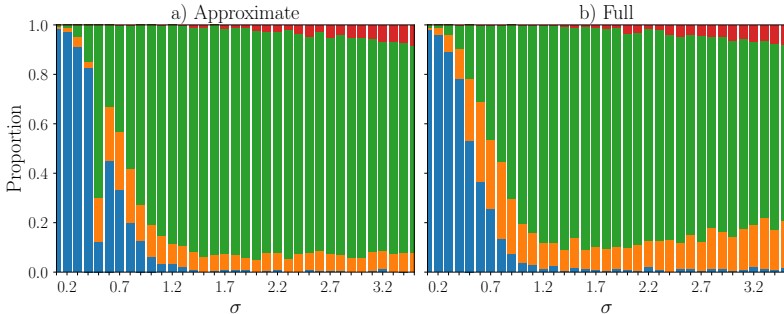

Figure 9: Proportion of correctly predicted instances for which each approach yields the highest certification across $\sigma$ for MNIST. Red represents the proportion for which the boundary treatment produces the largest certification, with Green, Orange, and Blue representing the same for Double transitivity, Single transitivity, or Cohen *et al.*. While our approaches subsume Cohen, if no other technique is able to improve upon the base certification, we assign the largest certification as having been calculated by Cohen *et al.*.

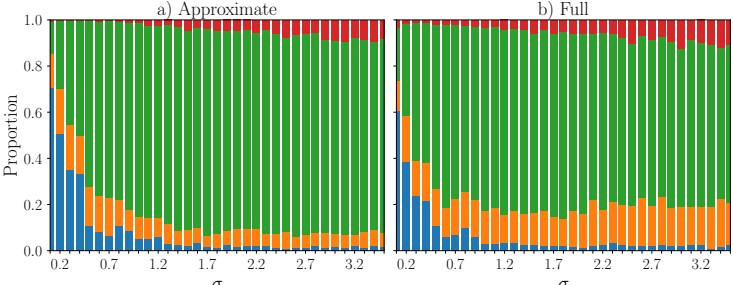

Figure 10: Proportion of correctly predicted instances for which each approach yields the highest certification across $\sigma$ for CIFAR-10. Red represents the proportion for which the boundary treatment produces the largest certification, with Green, Orange, and Blue representing the same for Double transitivity, Single transitivity, or Cohen *et al.*. While our approaches subsume Cohen, if no other technique is able to improve upon the base certification, we assign the largest certification as having been calculated by Cohen *et al.*.

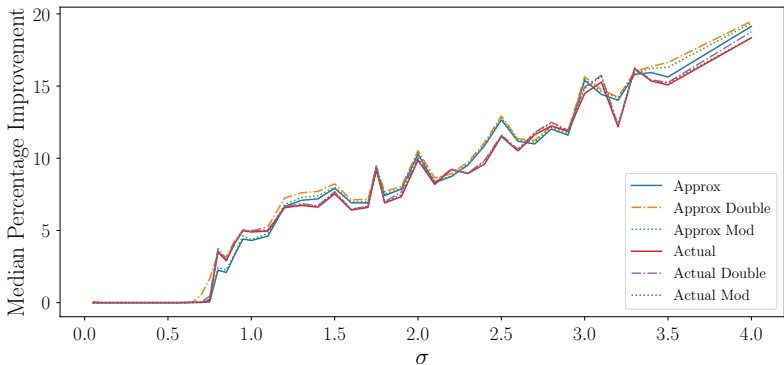

Figure 11: Median percentage improvement of the Certified Robustness achieved by each of our approaches relative to Cohen *et al.* for MNIST across the level of additive noise $\sigma$. The median was chosen to provide a fair and representative comparison to Cohen *et al.*, that filters out outliers in the percentage improvement when $r \ll 1$.

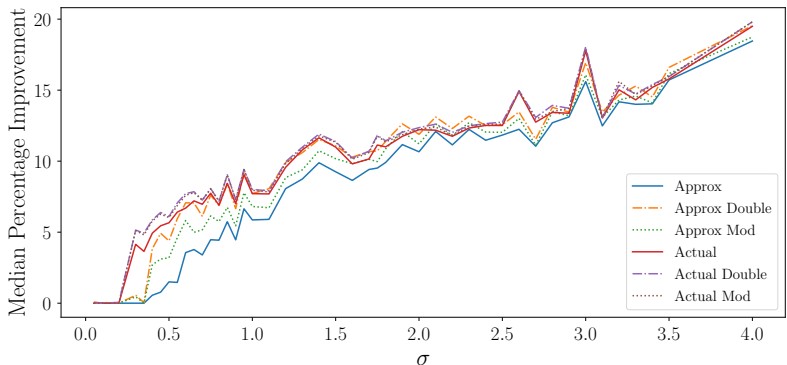

Figure 12: Median percentage improvement of the Certified Robustness achieved by each of our approaches relative to Cohen *et al.* for CIFAR-10 across the level of additive noise $\sigma$. The median was chosen to provide a fair and representative comparison to Cohen *et al.*, that filters out outliers in the percentage improvement when $r \ll 1$.

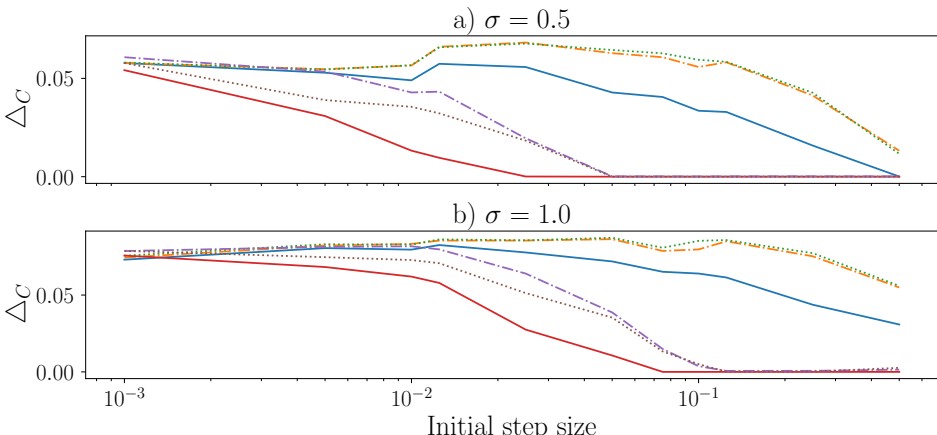

Figure 13: Average delta to the Cohen *et al.* Certified Radius ($\triangle_C$) for CIFAR-10 as a function of the initial step size $\gamma$. Here Blue, Green, and Yellow represent the Full Single transitivity, Double transitivity, and Boundary treatments; with Blue, Purple, and Brown representing the same for the Approximate solver.

### A.4    Influence of the starting step size

The one heretofore un-considered feature is the influence of the initial step-size $\gamma$ within Algorithm 1. As is shown in Figures 13 and 14, while the Full solver only exhibits sensitivity to $\gamma$ when $\gamma > 0.1$, the approximate solvers are far more sensitive, with deleterious performance being observed for $\gamma > 0.01$ in CIFAR-10, and even earlier for Tiny-Imagenet. This is likely due to the added uncertainty in the Approximate derivatives leading to convergence upon local sub-optima in the more semantically complex datasets. Based upon these results, the starting step size was uniformly set to 0.01 for all experiments.

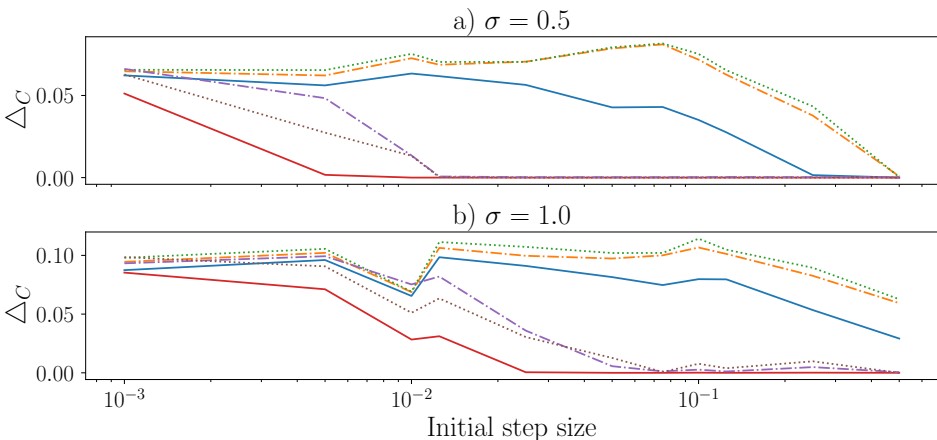

Figure 14: Average delta to the Cohen *et al.* Certified Radius ($\triangle_C$) for Tiny-Imagenet as a function of the initial step size $\gamma$. Here Blue, Green, and Yellow represent the Full Single transitivity, Double transitivity, and Boundary treatments; with Blue, Purple, and Brown representing the same for the Approximate solver.