# OpenReview forum: "Double Bubble, Toil and Trouble: Enhancing Certified Robustness through Transitivity"
_NeurIPS.cc/2022/Conference — NeurIPS 2022 Accept_

### Official Review · Reviewer_7Pxn · 2022-06-22

**Rating:** 5
**Confidence:** 3
**Soundness:** 2 fair
**Presentation:** 3 good
**Contribution:** 3 good

**Summary:**

The authors provide a method to enlarge the certification radii given by randomised smoothing based approaches. They use gradient based methods to find a points (and calculate their certification radii) in the proximity of the original input point, such that the combined information yields a larger certificate for the original input point. Two tricks are employed to be able to calculate the gradients yielding two methods: The ‘Full’ method, where the Gumble-Softmax is used to replace the argmax and the ‘Approximate’ method, which is based on reformulations and approximations.  The approaches are evaluate on MNIST, CIFAR-10, Tiny-Imagenet - the networks are based on Resnet18.

**Questions:**

- How would the improvements look like where more recent state of the art methods (as above) where to be used to train the network?
- What is the standard deviation between different training runs?
- Eq. 6: what does $r’$ refer to? The same as in Eq. 5?
- Lemma 3.4: Is Lemma 3.4 optimal? Is $x_3$ here now fixed or variable? In Eq. 7, it seems as if $d_3$ and $r_3$ depend on $x_3$, whereas in Eq. 9 and Eq. 10 there seems to be a (further?) dependence of $y$ on $x$.
- 152: To what does “feasible domain” refer to?
- Did the authors consider to evaluate on ImageNet?


**Limitations:**

The authors have mostly covered the limitations well. However, it remains unclear how scalable the method is to ImageNet.

**Strengths And Weaknesses:**

The authors tackle an important problem, as certified robustness is an important research field. The ideas are interesting and the contributions are to the best of my knowledge novel. In the evaluation section, the authors give good intuition when explaining their results. The ideas are general and seem (at least partially) applicable to approaches certifying the robustness of geometric perturbations.

The experimental evaluation seems weak in the sense that the authors compare to [5], but not showcase how much their method would improve the results on some of the more recent papers published in this direction including MACER (https://arxiv.org/pdf/2001.02378.pdf), SmoothAdv (https://arxiv.org/pdf/1906.04584.pdf) or Consistency Smoothing (https://arxiv.org/pdf/2006.04062.pdf). Further, Section 3.2 and 3.3 are difficult to follow. The proof of Lemma 3.4 is not well explained and could hence not be checked for correctness. This is likely due to some minor confusion that can be easily clarified by the authors. More in the questions. Formalisation and maybe some pictorial illustrations might improve this section. Similarly for Lemma 3.5. Also, it remained unclear how the method would perform on challenging datasets including ImageNet.

Regarding the related work: In line 52 and following, there is a misleading claim. The time and memory complexity are not the problem. For example interval bound propagation can be applied to analyse high dimensional inputs on large networks. The practical problem is the loss of precision linear relaxations suffer from. Further, the authors mention multiple times in the paper that “Certified robustness has not, however, extended to rotational or translational modifications …”. I encourage the authors to take a look at the following papers, the last 2 are randomised smoothing based:

https://arxiv.org/pdf/1712.01785.pdf

https://dl.acm.org/doi/10.1145/3290354

https://papers.nips.cc/paper/2019/file/f7fa6aca028e7ff4ef62d75ed025fe76-Paper.pdf

https://arxiv.org/pdf/1912.09533.pdf

https://arxiv.org/abs/2002.12463

https://arxiv.org/pdf/2002.12398.pdf

Minor: In line 121, “the” appears twice in a row and $x_j$ is typed differently.

***
After the authors response, i have increased the score.

---

> ### Author Response · Authors · 2022-07-29
> **Review Response**
>
> We offer our thanks to Reviewer 7Pxn for their comments, and their time.
>
> Regarding the comparison to papers like MACER, we believe that such papers are complimentary approaches, and are not directly comparable. This is because each technique utilises Cohen et al. as the underlying mechanism, before trying to improve the certification by modifying the training loop. In contrast, our work (and that of Cohen) does not require any modifications to the core training loop (in terms of loss functions or training routines), which we believe makes it more readily applicable to a broader range of problem spaces. By comparing against Cohen, we are making an argument for replacing Cohen as the central component of works like MACER.
>
> To support our contention that direct comparisons between our technique and MACER are not appropriate, we performed a baseline comparison for CIFAR-10 at $\sigma = 1.0$ for Resnet18 model, with $N=1,000$ employed for the uncertainty quantification. Under these conditions:
>
> || Train (s) / Epoch| Train Loss | % Cert | Outperform |
> |-------|-------------------|------------|--------|------------|
> | Ours|6|1.6|60 %|52 %|
> | MACER |40|2.15|16 %|15 %|
>
> We believe that this relative difference in performance is a result of MACER's hybrid objective, that balances decreases in the predictive losses against increases in the certified radii. This increases the training cost, and the higher achieved predictive training loss significantly reduces the number of samples that can be predicted accurately, and thus certified. We do note though that in the $16 \%$ of samples for which MACER is able to certify, it does produce certified radii that are on average $2.34$ larger than our technique.
>
> Relating to the point on Line 52 - line 53 cites 'Beta-CROWN' by Wang et al (which we believe is the most recent paper on IBP) which states that "5,400 GPU hours...to verify 200 testing images" for a CIFAR-$10$ dataset using Resnet5. That this level of time is required for CIFAR-$10$ suggests to us that scaling to datasets with larger images would be infeasible. However, we acknowledge that this is our interpretation and have made this clear within the Rebuttal Revision. While the reviewer has cited several papers, we believe that only `Certifying Geometric Robustness of Neural Networks' utilises bound propagation as described in the section about Line 52, and we note that this paper also appears to only present results up to CIFAR-$10$.
>
> Relating to translation / rotation, we thank the reviewer for these citations and have incorporated them within the Rebuttal Revision.
>
> Regarding your questions:
>
> 1) Our experimental results on MACER suggest that our technique still yields improvements in the certified robustness, while requiring significantly less training time.
>
> 2) Our experiments have suggested that there is relatively insignificant variance across training runs. This is a product of each expectation being calculated over an average of $1,000$ Monte Carlo samples, with certifications being taken based upon high probability lower-bound of said expectations. As such, the net variance is very small, although to ensure repeatability our final results used a fixed random seed.
>
> 3) Yes, $r'$ in Equation 6 is the result calculated in Equation 5, which has been clarified in the rebuttal revisions.
>
> 4) Lemma 3.4 is optimal if Definition 3.3 is true - ie. if $x_3$ sit on the vector spanning $x_1$ and $x_2$, then the achieved certified radius is calculable as per Lemma 3.4. While it is not guaranteed that the optimal $x_3$ sits upon this vector, doing so would produce a symmetry breaking event that would require any such point off the vector to have a significantly larger radii of certification than a point on the vector. As changes in the size of the radii of certification are bounded, we assume that positioning $x_3$ following Definition 3.3 and Equation 6 is close to optimal (and in doing so we also significantly reduce the search complexity), and if that is true then Lemma 3.4 is a provably true.
>
> 5) Following Line 151 - x is restricted to $[0,1]^d$, and the feasible domain on Line 152 refers to this. We have added additional clarification to the Rebuttal Revision.
>
> 6) The authors did consider testing against Imagenet, however computational limitations prevented us from being able to properly do so at this point in time. As such, we utilised Tiny-Imagenet, which is a 200-class, 3x60x60 sampling of Imagenet to demonstrate the performance of our technique upon larger, more semantically complex datasets. We note that Tiny-Imagenet images are still significantly larger than other datasets that are commonly tested (like CIFAR-$10$) and that our relative performance increases over the baseline measure of Cohen appears to be broadly unaffected by changes in the dataset size, and as such we are confident that there is no intrinsic property of the proposed techniques that would fail to scale towards larger datasets.

---

> > ### Comment · Reviewer_7Pxn · 2022-08-08
> > **Follow up**
> >
> > Thank you very much for your response. As most of my concerns have been addressed by the authors, i will increase my score. Some more comments:
> >
> > Regarding the related work:
> > - It seems as if the authors seem to confuse some details. IBP is short for Interval Bound propagation ([see paper](https://arxiv.org/abs/1810.12715)). Beta-CROWN does not propagate intervals through the network, hence it is not IBP. Both, IBP and Beta-CROWN are instantiations of convex relaxations though.
> > - Further "An abstract domain for certifying neural networks" uses convex relaxations (used essentially as a baseline in "Certifying Geometric Robustness of Neural Networks"). Similarly "Towards Verifying Robustness of Neural Networks Against A Family of Semantic Perturbations" also relies on relaxations.
> >
> > Lemma 3.3:
> > - I think i am missing some minor detail here: do i suppose correctly that $[0,1]^d$ is the entire input space? Why exists some $(x_3, r_3)$ such that $B_P(x_3, r_3) \not\subset B_P(\tilde{x}, \tilde{r})$ $\forall \tilde{x} \in [0,1]^d$, when we can pick $\tilde{x} = x_3$? It appears to me as if in this case $\tilde{r}$ (-> Eq. 9) would coincide with $r_3$.
> >
> > I think it would make sense for the authors to include the current limitation regarding the scalability to large datasets in the limitations section.

---

> > > ### Author Response · Authors · 2022-08-09
> > > **Thanks + Response**
> > >
> > > We thank the reviewer for both their comments, and their updated score.
> > >
> > > > Relating to IBP
> > >
> > > The reviewer is correct that we were thinking more Convex Relaxation based methods in our response (as that's what we primarily cite in the paper) rather than the Interval Bound Propagation that you mentioned, and we should have been a bit more careful in our answer there. However, our response was motivated by our belief that hybrid Convex Relaxation/IBP approaches were becoming the defacto implementation of IBP (although we acknowledge we may be wrong on this front), as they're guaranteed to give tighter bounds than IBP [1,2], and do not appear to exhibit the same training instability as IBP [2]. Similar to those issues exhibited by Convex Relaxation, multiple works have noted scalability issues in IBP when scaling towards ImageNet [3,4], with those papers that do reach this scale either using multiple TPUs with unspecified ram on downscaled ImageNet (equivalent to TinyImagenet) [5], or noting significant training times and resource demands for CIFAR-10 and then not specifying training time or resources used for ImageNet [6].
> > >
> > > To add depth to our coverage of bound based techniques, we've attempted to broaden the section on Bounding Mechanisms (Lines 43-50), including incorporating additional citations.
> > >
> > > > Lemma 3.3
> > >
> > > You're correct that [0,1]^d is the entire input space, and that you could pick that $\tilde{x}$ could coincide with $x_{3}$. To account for this we've modified Equation 9 to put an exclusionary condition upon $\tilde{x}$.
> > >
> > > > Limitations
> > >
> > > We've modified lines 258-269 in the limitations section to more explicitly consider the time and memory complexity of our approach, relative to standard randomised smoothing robustness mechanisms, and have incorporated a discussion of potentially improvements in the memory complexity.
> > >
> > > We hope that these comments have fully addressed your concerns, and thank you again for your willingness to provide additional comments.
> > >
> > > > References
> > >
> > > [1] "Towards Evaluating and Training Verifiably Robust Neural Networks" Lyu, Z. et al., CVPR (2021)
> > >
> > > [2] "Automatic Perturbation Analysis for Scalable Certified Robustness and Beyond" Xu, K. et al., NeurIPS (2020)
> > >
> > > [3] "Certified Defences for Adversarial Patches", Chiang, P-y., et al., ICLR (2020)
> > >
> > > [4] "Efficient Certification of Spatial Robustness", Ruoss, A. et al., AAAI (2021)
> > >
> > > [5] "On the Effectiveness of Interval Bound Propagation for Training Verifiably Robust Models", Gowal, et al., arXiv (2019) [Originally published NeurIPS, 2018]
> > >
> > > [6] "Fast, Certified Robust Training with Short Warmup" Shi, Z., et al., NeurIPS (2021)

---

> ### Author Response · Authors · 2022-08-08
> **Additional response**
>
> Just as an update, we would just like to emphasise that subsequent to your review we have updated the paper in response to both your comments, and those of the other reviewers. Specifically relating to your comments, these updates have included clarifying what were Equations 5/6 and Lemma 3.4, and the comments relating to certifications against rotational/translational attacks. To assist in assessing these changes, the final pages of the Rebuttal Revision have the changes to the document highlighted in red/blue for deletions/additions.
>
> We also wish to highlight that our other rebuttal comment outlines the difference between our approach and that taken by MACER (and other similar approaches), as requested by you in your original review. In contrast to MACER, which requires significant updates to the training loop, a decrease in both accuracy and a significant increase in training time, our approach can be applied to general model training architectures, with a much smaller computational impost, while still significantly out-certifying MACER on a like-for-like comparison. That our reference point was Cohen et.al was deliberate, in that both techniques are able to be generalised simply, and both have the potential to sit inside augmentation approaches like MACER (which uses Cohen as its core certification routine).
>
> Thanks again for your time and your consideration

---

### Official Review · Reviewer_R63M · 2022-06-23

**Rating:** 6
**Confidence:** 4
**Soundness:** 4 excellent
**Presentation:** 2 fair
**Contribution:** 3 good

**Summary:**

The paper presents geometrically-informed certified robustness to improve the certified radii computed by randomized smoothing techniques. The paper presents three approaches for geometrically-informed certified robustness: transitivity, multiple (double) transitivity, and boundary treatment. The evaluation shows that the proposed approaches equipped with two optimization algorithms can outperform the baseline.

**Questions:**

1. The paper mentions "exponential growth in the number of hyperspheres that need to be solved" on lines 146-147. Then how about the results with small n like $n=4,5,6$?
2. Do the three approaches subsume Cohen et al.? If they do, please state the claim in the main paper and why Figure 4. has some cases won by Cohen et al.? If they do not, please discuss the reason or provide the counterexample.
3. What is $r_1$ in Figure 3.
4. What are the "Approx", "Approx Mod", "Actual", and "Actual Mod" in Figure 5?
5. What is the choice of the iteration $M$ across all experiments?

**Limitations:**

The paper only presents the geometrically informed certified robustness on the l_2 norm as stated in section 5. However, it is encouraged for authors to discuss other useful norms such as the l_1 and l_inf norms. Specifically, it is not easy to extend the multiple transitivities to other norms.
Algorithm 1 needs M iterations, each requiring thousands of Monte-Carlo sampling.

**Strengths And Weaknesses:**

Overall, it is a paper with a novel and effective idea but needs improvement in writing.

Strength:
The paper presents geometrically-informed certified robustness to improve the certified radii computed by randomized smoothing techniques. The paper identifies a hard problem and proposes simple yet effective approaches to tackle the problem. I like the idea of geometrically-informed certified robustness because the bound computed by randomized smoothing techniques is tight under assumptions that they over-approximated the worst-case behavior of the machine learning classifier **only** with respect to the constraints of E0[x] and E1[x]. In other words, other approaches ignore the underlying classifier's complex behavior (geometry). However, this paper uses additional information by choosing $x'$ with $E0[x']$ and $E1[x']$, i.e., the geometry around $x'$ to improve the certification. Ideally, if we choose a finite set of $x'$, we could approximate the geometry around $x$, yet this approximation is expensive and requires exponential computation.

Weakness:
- The writing in sections 3 and 4 should be greatly improved:
  1. I appreciate that the paper provides three figures for illustration. However, it is important to clarify these circles, i.e., mark out x1, x2, x3. It would be better to show the computation of these 2D examples.
  2. Define notation before usage and the inconsistency in notations:
    * $r_2(x_2)$ in Eq 5;
    * $x_3(s)$, $s$, and $r'$ in Eq 6;
    * $j$ and $k$ in Eq 8 (Here, $j$ should be $\forall j$ and $k$ is a special dimension);
    * $\hat{r_k}$ in Eq 13. What are $x_3', r_3'$ in definition 3.3?
    * The notation of $y_{2,k}$ appears in the proof of Lemma 3.4, yet the definition appears later in the proof of Lemma 3.5. Moreover, the definition later of $x_n$ is wrong, should be $\{x_{n,1}, \ldots, x_{n,d}\}$.
    * At the beginning of the proof of Lemma 3.5 until Eq 12, $x_i$ should be $z_i$.
    * $c$, $\lambda$, $\Gamma$ are not used in Algorithm 1.
    * $i$ should be $k$ in the second line of Eq 15.
  3. State the intuition of Eq 6. Maybe say x_1, x_2, x_3 should be in the same $R^{d-1}$ hyperplane?
  4. In the proof of Lemma 3.4, mention that $\mathbf{y}_1$ is at the origin after translation and rotation.
- The paper only presents the l_2 norm case as stated in section 5. However, it seems not easy to extend the multiple transitivities to other norms, e.g., l_1 and l_inf.

Other comments:
1. Missing related works:
  * The paper states, "Certified robustness has not, however, extended to rotational or translational modifications" on lines 68-69. However, there are already works [1,2] that handle the 2D and 3D image transformations.
  * Another recent paper [3] presents a different idea to improve the certified radius of randomized smoothing by incorporating another distribution.
2. It is interesting to explore the combination of the geometrically-informed certified robustness with other training methods [4,5] like [3] does.

[1] Linyi Li, Maurice Weber, Xiaojun Xu, Luka Rimanic, Bhavya Kailkhura, Tao Xie, Ce Zhang, Bo Li, TSS: Transformation-Specific Smoothing for Robustness Certification

[2] Wenda Chu, Linyi Li, Bo Li. TPC: Transformation-Specific Smoothing for Point Cloud Models

[3] Linyi Li, Jiawei Zhang, Tao Xie, Bo Li, Double Sampling Randomized Smoothing

[4] Jeong, J. and Shin, J. Consistency regularization for certified robustness of smoothed classifiers

[5] Jeong, J., Park, S., Kim, M., Lee, H., Kim, D., and Shin, J. Smoothmix: Training confidence-calibrated smoothed classifiers for certified robustness.

---

> ### Author Response · Authors · 2022-07-29
> **Review response**
>
> We offer our thanks to reviewer R63M for their responses, and the detailed corrections they have offered, which have been accounted for in the Rebuttal Revision. To respond to the questions raised
>
> 1) The growth in complexity is primarily driven by the need to ensure coverage across the $R^d$ space. To clarify this point (which we have attempted to address in the rebuttal revisions), with one additional hypersphere, covering the original certification region is simple. If we are then to add a second hypersphere, the optimal positioning for this hypersphere is likely to sit upon the vector spanning $x_1$ and $x_2$, on the other side of the ball to $x_1$, as the nearest point to $x_1$ on the ball about $x_2$ is a point on this vector. However, if we were to then seek to add a fourth hypersphere, we face the problem that the nearest point to $x_1$ on the union of the two prior hyperspheres is a $d-1$ dimensional manifold in $R^d$ space, which we would need to completely cover with an exponentially larger number of hyperspheres.
>
> If we were to abandon the process of iteratively adding one hypersphere at a time, then there may be a configuration of $n=3$ hyperspheres that would produce a further increase in the certified radius. However, we note that searching for such a set would require a significant increase in the computational complexity of the search process, and the achievable gains would likely be vanishingly small with increasing $n$. Given the reviewers questions on this point, we have taken the opportunity to clarify our position within the Rebuttal Revision.
>
> 2)  It is correct that this approach subsumes Cohen et al. The Figure 4 caption has been be improved to make it clear that we consider Cohen to outperform when no other technique is able to yield an improved certification, in that the additional computational complexity of the other approaches has yielded no material improvement.
>
> 3) Regarding Figure 3, this figure presents the average improvement for samples within $\pm 0.75$ of a given $r$, in order to separate out if the influence of our technique is highest for small or large radii certifications - the explanation for this has been revised in the Rebuttal Revision.
>
> 4) The Approx / Actual demarcation should be Approx / Full (as per lines 193-195). The three demarcations also should be Single (currently just labelled as Approx / Actual), Double, or the Boundary Treatment (currently labelled with Mod). Unfortunately the labelling in this Figure is from a draft version of the paper, and should have been corrected prior to submission. While we are unable to correct this labelling for the Rebuttal Revision (as the machine on which we perform experiments has been temporarily decommissioned at a most inopportune time), it would be corrected in the final version of this paper.
>
> 5) All experimental results were based upon $1000$ randomly drawn samples from each dataset.
>
> Relating to the identified limitation, our work specifically exists within $L_2$, as it is the only space in which there currently exists a guaranteed best-case certification (Cohen et al). However, it should be true that other $L_P$ norms also produce certifications that exist as bounding hyperspheres (we presently have preliminary work on this for $L_1$ norms, that we are looking to generalise to $L_P$ space), and if this contention holds then the multiple transitivity approach should also be extensible to these $L_P$ spaces.
>
> Finally, to adress the level of Monte-Carlo sampling required - while we do have a plan to reduce the sample complexity of these approaches, we consider this work to be a proof of concept that will be reinforced by optimisation of the numerical process at a later stage. Additional detail regarding this has been added to the Limitations section of the Rebuttal Response.

---

> > ### Comment · Reviewer_R63M · 2022-08-02
> > **Reviewer-author Discussion**
> >
> > Thanks for the response. I find the response well addressed most of my comments. I also looked at the revised version. I understand that it is a short revision period, so some of my points were not addressed:
> >
> > 1. I appreciate that the paper provides three figures for illustration. However, it is important to clarify these circles, i.e., mark out x1, x2, x3. It would be better to show the computation of these 2D examples.
> > 2. $r_2(x_2)$ in Eq 5 is not defined.
> >
> > I will keep my rating at this time.

---

> > > ### Author Response · Authors · 2022-08-04
> > > **Response with updated rebuttal revision**
> > >
> > > We thank you for pointing those two elements out. While they were missed in the original revisions we are now confident that we have addressed both points in our latest uploaded variant.
> > >
> > > We hope that our updates have resolved all your concerns regarding the paper, and would welcome any additional questions.

---

### Official Review · Reviewer_diuY · 2022-07-08

**Rating:** 7
**Confidence:** 3
**Soundness:** 3 good
**Presentation:** 4 excellent
**Contribution:** 4 excellent

**Summary:**

This paper proposes to exploit the certified robustness of samples in the neighbourhood of a given data in order to enhance the certifiable radius of the given data. To be specific, the authors leverage the intersection of the certified regions of multiple neighbourhoods. Compared to the previous state-of-the-art method, the experimental results validate the effectiveness of the proposed method in enhancing certified robustness.

**Questions:**

- Could the authors provide any insights on how to accelerate the proposed method?
- I am a bit confused about the meaning of some part of the title — ‘Double Bubble, Toil and Trouble’. Could the authors provide some explanations about this part?


**Limitations:**

The authors have pointed out the limitations in details in the main paper.


**Strengths And Weaknesses:**

Strengths
+ The proposed method is interesting and novel. The method is (arguably) supported by theoretical analyses.
+ The authors conduct comprehensive experiments. The results on various datasets, different hyperparameters, and different transitivity approaches all support the claim that the proposed method can improve certified robustness.

Weaknesses
- It seems that the proposed method could be somewhat more time-consuming compared to previous state-of-theart method.

---

> ### Author Response · Authors · 2022-07-29
> **Review Response**
>
> We thank the reviewer for their time and comments. Relating to your question regarding accelerating the proposed method, our plans for subsequent numerical optimisation involve incorporating multiple importance sampling, to significantly reduce the number of model evaluations that are required to produce the certifications. As the model evaluations are the key driver of the computational cost, any such decreases should significantly accelerate the certification process, and we have added a note relating to this within the Limitations section in the Rebuttal Revision.
>
> We emphasise that while the proposed technique (without numerical optimisation) does increase the time for certification, the ability to produce larger guarantees would be a more than acceptable as a trade off in high-value systems considered vulnerable to adversarial attacks.
>
> Relating to the title, the title is a pun based upon a line from the Shakespeare play Macbeth which states “Double, Double, Toil and Trouble”, which we felt was appropriate as the technique uses two certification hyperspheres (or bubbles) to certify against troubling adversarial interactions. Perhaps it was a pun too far for the title.

---

### Official Review · Reviewer_JQxG · 2022-07-10

**Rating:** 6
**Confidence:** 4
**Soundness:** 3 good
**Presentation:** 2 fair
**Contribution:** 4 excellent

**Summary:**

This paper addresses the problem of designing local robustness certification procedures for neural classifiers. The primary insight of the paper is that given an input x, instead of simply calculating the radius of the robust ball B centered at x, one should also consider calculating robustness radii at other points in the ball B. It is easy to see that the network is robust in the entire region defined by the union of B and the robust balls centered at these additional points. Moreover, if such additional points are suitably chosen, one can achieve a larger robustness radius at x.
The paper, in particular, focuses on local robustness certification via Randomized Smoothing. In order to choose suitable additional points for robustness certification, the authors formulate optimization problems for choosing up to two additional points to certify. These optimization problems are solved at inference time to choose the additional points to certify. Since the robustness radius calculation used by Randomized Smoothing requires computing expectations, special care is needed when solving these optimization problems via gradient descent. The paper presents two different approaches for solving the optimization problem so that the required gradients can be calculated.
Empirical evaluation is performed using MNIST, CIFAR-10, and Tiny-Imagenet datasets. The evaluation suggests that the proposed technique is indeed able to produce larger certifiable balls and improve upon the certified accuracy of randomized smoothing.

**Questions:**

Apart from the questions above, below are some more questions and comments:
(1) Line 72,73 says that the certified radius formula (Equation 3) of Cohen at al. gives a lower bound on the certified radius. On the other hand, lines 36, 91, 95 etc state that this is the largest certifiable radius. Is it a lower bound or a tight bound? I believe its the latter but please provide clarification

(2) Line 75: runner -> runner-up

(3) Line 79: binomial -> binary?

(4) Defn 3.1 seems incomplete. First, what is the definition of B_P(x1,r1), B_P(x2,r2)? Second, is it not the case that x1 and x2 have to be in each other's certifiable regions for the definition to make sense? To me, it seems like Defn 3.1 is not really a definition, but it needs to be a theorem with a proof.

(5) Line 114: recast the certification problem -> recast the certification radius

(6) Line 123: r_n seems overloaded since its already used to denote the certified radius for point x_n. Should r_n instead be r''' or something?

(7) Line 122 and Defn 3.3: Questions about the meaning of S_n, s, and various questions about Defn 3.3 are mentioned above under "weaknesses of the paper".

(8) Proof of Lemma 3.4: Questions about the proof are mentioned under "weaknesses of the paper".

(9) Line 151: Why was the notation switched from r1 to L1?

(10) Line 164: x_n is defined as {x_1,n, x_2,n, ...}. On the other hand, in Equation (8), in y_2,j y_3,j, the j'th coordinate of points y2 and y3 are being referred. Please make the subscript order consistent.

(11) Questions about Algorithm 1 have already been mentioned earlier

(12) Equation 14: i should be replaced by k

(13) Equation 15: All the i's need to be replaced by k

(14) Line 225: How does increase in R affect the value of E1? Isn't E1 independent of R?

(15) Questions about Figure 2, 3, and 5 have already been mentioned earlier.

(16) In Table 1 reporting the average amount of time required for certifying a single input? How many samples are drawn while reporting the results for Cohen et al.? Is it the same number of samples as for the other approaches, i.e., 1000 samples?

(17) Cohen at al. already introduces significant overheads (due to sampling and evaluating the model on thousands of inputs). Would it not be a problem to add their overheads by one to two orders of magnitude as shown in Table 1?

**Limitations:**

The authors describe some limitations of their approach in Section 5 under the para labeled "Limitations". It would be nice to have some more discussion about the overheads introduced by the approach.

**Strengths And Weaknesses:**

Strengths of the paper:
(1) The proposed idea to increase the size of the certified balls is original yet intuitive. Its a broadly applicable idea that can be applied to other local robustness certification procedures, apart from Randomized Smoothing.

(2) The empirical results provide compelling evidence that the idea could lead to certification gains in practice.

Weaknesses of the paper:
Though I really like the idea presented in the paper, the presentation and quality of writing in the paper needs significant improvement before the paper can be published. I list my main concerns below:

(1) The mathematical writing in the paper is very hard to follow. Often, the symbols are either not defined or not used in a careful manner. I could barely follow Sections 3.2 and 3.3. Though I believe I have a sense of what the authors are trying to say, the actual technical definitions and proofs were not helpful in understanding the material. For instance, consider Defn 3.3. I have no idea what its trying to say. What are s and r' there? Is s a vector quantity? If yes, then the rightmost term in Equation 6 involves a multiplication of two vector quantities. Is it supposed to be a dot product of two vectors? What is s anyway? Why is it of dimension R^(d-1) when the inputs are of dimension d? Line 128 says, "assuming the above definition is true". What does this mean? How can a definition be false? Does Defn 3.3 need to be a theorem instead? Similarly, consider the proof of Lemma 3.4. What are y_2 and y_3? What is k in equation (8)? I couldn't follow the proof at all and have no idea if Lemma 3.4 is indeed true or not.

(2) Related to the above point, even the descriptions of the mathematical statements are hard to follow. For instance, consider the first two paras of Section 3.3. Even after reading them multiple times, I was not able to figure what the authors were trying to say. My understanding is that the boundary treatments can sometimes allow larger robustness radius because the closest point on the red circle (referring to Figure 1) is outside the feasible bounded region, so one can consider points on the red circle that are farther away as being closest to the test input. However, the text in Section 3.3 did not help me in figuring this out. It would be very helpful, for instance, to explain the situation using the already existing Figure 1.

(4) Algorithm 1 needs to be more carefully written. Where are the input parameters c, \lambda, \Gamma used? Is Algorithm 2 called on line 3? Why not write it as Algorithm2(x',N,\sigma) instead of Model(x + ....)? In fact, writing it as Model(x + ....) is incorrect since line 3 is not executing the model at a particular input but instead executing Algorithm 2. There is no description of how step size \gamma is calculated on line 9 even though a paper is cited. And what is the justification for line 11? It would help to have a more detailed description of Algorithm 1 in Section 3.4.

(5) The presentation of the empirical results is also quite confusing. Although a lot of detail is provided, important high-level details are missing. For instance, I was not able to figure out how to interpret Figure 3. What does the x-axis show? What does Figure 5 report? Percentage Improvement in what quantity? Is it reporting percentage improvement in "certified proportion"? Also, why is y-axis of Figure 2 labeled "certified proportion"? Cohen et al. call it "certified accuracy" which is a much more informative name. If Figure 5 is indeed reporting median percentage improvement in certified accuracy, at what radius is the certified accuracy being calculated?

===============================
Post one round of rebuttal:
Changed *Contribution* from 3 to 4
Changed *Rating* from 3 to 4

Post second round of rebuttal:
Changed *Presentation* from 1 to 2
Changed *Rating* from 4 to 6

---

> ### Author Response · Authors · 2022-07-29
> **Review response**
>
> We thank reviewer JQxG for their time and expertise. Please note that our new Rebuttal Revision attempts to address the majority of points raised. We would like to to respond to a few specific points here, beginning with itemised weaknesses:
>
> * (1) We've clarified Definition 3.3 in text. If a third hypersphere is added, then the centre of $x_3$ should lie on the vector spanning $x_1$ and $x_{2}$. This assumption greatly simplifies the search space, and intuitively the optimal location of $x_3$ should be close to this line (but it is not guaranteed true that the optimal location is on it, thus the comment in Line 128). A sphere on this line will have symmetric intersections with the ball about $x_2$.\
> \
> Regarding Lemma 3.4's proof: In the text we note on line 134 that $y_2$ and $y_3$ are a coordinate mapping of $x_2$ and $x_3$ so that only one component of $y_2$ and $y_3$ are non-zero&mdash;a mapping that we are free to do from the symmetry of the underlying hyperspheres, and which dramatically simplifies the proof. This point, and the prior one, have both inspired updates to the paper, available as the Rebuttal Revision.\
> \
> Relating to the the comment about dimensionality&mdash;while the surface of the hyperspheres is $R^d$, the surface of any boundary of intersection is a $d-1$ dimensional manifold embedded in $R^{d}$ (as noted in Line 144). Thus adding any balls after $x_3$ has been established as practically infeasible, as it involves guaranteeing coverage across $R^{d-1}$. This is clarified within the Rebuttal Revision.
>
> * (5) Figure 3 presents the average improvement for samples within $\pm 0.75$ of a given $r$, in order to distinguish whether the influence of our technique is highest for small or large radius certifications&mdash;this explanation has been revised in the Rebuttal Revision. Figure 5 presents the influence of the noise level $\sigma$ upon the average improvements (across all samples) that are possible through our revised approaches. We agree regarding Figure 2, and while we are unable to revise it for the Rebuttal Revision (due to a recently decommissioned computer), a finalised version of this paper would unify the language with Cohen *et al.*
>
> Relating to the itemised questions, while most of these are regrettable typos that we hope to have addressed in the Rebuttal Revision, we would like to respond to selected key points:
>
> * (1) Lines 72&ndash;73 state that Cohen provides a “lower bound guarantee of class invariance”, in that Cohen produces a guarantee that the class will not change for perturbations up to the size of the certification, under the assumption that the decision boundary is the worst possible decision boundary for the purposes of considering sensitivity to perturbations. Lines 36, 91, 95 refer to the fact that under this worst possible boundary, the certification provided by Cohen is provably the largest possible certified radius that can be calculated based upon the class expectations at a sample point.
>
> * (14) While $E_1$ is independent of $R$, $R$ depends on $E_1$, although for any given $R$ there are multiple potential values of $E_1$. Line 225 refers to the relative performance of the tested techniques as $E_1$ varies.
>
> * (16) Yes, all numbers of model draws are fixed and constant.
>
> * (17) While not every system requires certification (or enhanced certification relative to Cohen), there will be high-value systems for which improving the radii of certification is a more than acceptable trade-off for decreased model throughput. This also relates to the point raised under limitations. To both points: this work primarily exists to demonstrate the validity of using a geometric perspective to improve the radii of certification. We believe this conceptional contribution is valuable for certification research broadly. We hope to see follow up work deriving numerical optimisations. We have added additional detail regarding the computational cost within the limitations section.

---

> > ### Comment · Reviewer_JQxG · 2022-08-04
> > **Response**
> >
> > I thank the authors for their response and for updating the submitted paper. The changes in Section 3.2 and 3.3 are helpful and I was able to make much more sense of the material. However, the presentation still needs improvement. I have increased my rating by 1 point.
> >
> > 1. Perhaps this is due to my lack of expertise but I still don't understand why $\mathbf{s}$ in $\mathcal{S}_n(\mathbf{s})$ is $d-1$ dimensional? Lets say $n=1$. In this case $\mathcal{S}_n({\mathbf{s}})$ is just the $d$ dimensional hypersphere, right? What does $\mathbf{s}$ represent in this $n=1$ example?
> >
> > 2. Definition 3.3 should instead be Lemma 3.3 which proves that "if the condition on lines 120-121 holds true then the position of $x_3$ is given by Equation (6)". Then, for the derivation of Lemma 3.4, one needs to assume that the condition on lines 120-121 holds.
> >
> > 3. I don't understand what the condition on lines 120-121 is saying. What do you mean by auxiliary point $\tilde{\mathbf{x}}$?
> >
> > 4. Should "boundary hypersphere" on line 137 be "exterior boundary" instead? By boundary hypersphere, do you mean $\mathcal{S}_n(\mathbf{s})$? If yes, it would help to clarify in the paper.
> >
> > 5. The idea explained on line 137 that, at n=3, there are multiple closest points and they constitute a $(d-1)$ dimensional manifold should be explained more slowly, perhaps with the aid of Figure 1.
> >
> > 6. Is there formal evidence for saying that the complexity scales non-linearly with d (lines 139-142)? If not, then perhaps its best to avoid making the claim.
> >
> > 7. Line 3 of Algorithm 1 still has typos. It should be $\text{Algorithm 2}(x + \mathcal{N}(0,\sigma^2\mathcal{I}_N), N, \sigma)$
> >
> > 8. Do Figures 3 and 5 report percentage improvement in the certified radius? If yes, *please* say this in the captions of Figure 3 and 5.
> >
> > 9. Why are you using average in Figure 3 and median in Figure 5?

---

> > > ### Author Response · Authors · 2022-08-05
> > > **Response with reference to updated paper**
> > >
> > > Thank you for your updates score and the additional comments, which we hope will allow us to further clarify elements of the paper.
> > >
> > > 1: S_n(s) is the d-dimensional surface of the hypersphere, however, the closest point to $x_1$ upon the surface of intersection between $\{S_{2}, \ldots, S_{n-1} \}$ is a $d-1$-dimensional manifold. In the case of the 2D exemplar of Figure 1b), this can be seen in the equidistant intersections between the circles about $x_{2}$ and $x_{3}$. In higher dimensions, these equidistant points of intersection $S_{2} \cap S_{3}$ expands from two points to a $d-1$-dimensional manifold (specifically, a $d-1$-dimensional hypersphere if Lemma 3.3 holds) embedded in $\mathcal{R}^d$.
> > >
> > > We hope this helps explain the process here, and have also added additional content to Section 3.2 in order to ensure that this point is clear within the paper.
> > >
> > > 2: The reviewers point is well made about Definition 3.3, and we have changed this to a Lemma, as well as attempting to rewrite both Lemma's 3.3 and 3.4 to improve clarity.
> > >
> > > 3: This was meant to refer to some additional point $\tilde{x}$, to present a condition for Definition/Lemma 3.3 to be optimal. We've attempted to clear up the framing around this in the Section 3.2 in our latest update to the paper.
> > >
> > > 4: The aim of this paragraph was to refer to the intersection between the boundary hyperspheres, and we acknowledge that our attempts at clarifying could have been made clearer. In response to both this point and Point 1), we've revised this paragraph, and we hope that the updated definition more clearly conveys the nature of these surfaces.
> > >
> > > 5., 7., 8: We agree with all these points, and have made corrections in response to these points.
> > >
> > > 6: The reviewer is correct that this was a statement from intuition, rather than having any direct evidence. However, motivated by this comment we've both revised the statement, and *added Appendix A.2 which demonstrates that a lower bound on the number of bounding spheres required exhibits exponential growth*.
> > >
> > > 9: Figure 5 employs the median to reflect the likely improvement achieved by a sample, as the average percentage improvement is significantly distorted by samples for which the initial certification was $\ll 1$. Taking average across $\sigma$ produced results that were significantly more favourable to our approach, however we felt presenting them as such presented an uninformative picture of the dependency upon $\sigma$, and more broadly formed an unfair comparison to Cohen et. al. To help clarify this, we've updated the caption for Figure 5.
> > >
> > > In contrast, the rolling rolling window (taking the median over a range of $\pm 0.075$) of Figure 3 ensures that the average percentage improvement at any $r$ was not distorted by the samples for which $r \ll 1$. Because the number of samples within these $r \pm 0.075$ windows is generally small, the difference between the median and average percentage increases would be small, and we would be willing to revise this figure for the final submission of this paper.
> > >
> > > We again offer our thanks, and hope that our revisions have helped refine the presentation of this paper - the changes for which can be seen in our updated rebuttal response (which has an annotated diff at end of the document).  If there are any other concerns relating to your perception of our presentation, we would be happy to discuss these points further.

---

> > > > ### Comment · Reviewer_JQxG · 2022-08-05
> > > > **Response**
> > > >
> > > > I thank the authors for all their updates to the paper. I am much more happy with the presentation and have updated my rating to 6. I have to admit though that I am still a little worried that there might be other lingering presentation issues and typos that haven't been noticed by the reviewers and I would strongly urge the authors to make a careful pass over the paper.
> > > > I have a few more comments:
> > > >
> > > > 1. I understand the authors' comments about how the closest points to $x_1$ form a $d-1$ dimensional manifold. However, I am sorry but I still don't understand the statement on lines 136-137 of the revised paper. As per these lines, $\mathcal{S}_n(\mathbf{s})$ is the surface of the union of $d$-dimensional hyperspheres $B_P(r_i,x_i)$. So, for instance, in Figure 1(b), $\mathcal{S}_3(\mathbf{s})$ is the non-circular curve representing the exterior boundary of the union of the three circles. What does this have to do with the fact that closest points to $x_1$ form a $d-1$ dimensional manifold? It's still not clear to me why $\mathbf{s}$ is $d-1$ dimensional. Is this the standard way of defining a surface / exterior boundary?
> > > >
> > > > 2. Similar to the comment above, I think lines 176-186 have typos. Wherever you use $\mathcal{S}_2$ and $\mathcal{S}_3$, I think you mean to say surface of $B_P(r_2,x_2)$ and $B_P(r_3,x_3)$. As per the comment on lines 136-137, $\mathcal{S}_3$ represents exterior boundary of the union of the 3 hyperspheres and *NOT* the surface of $B_P(r_3,x_3)$.
> > > >
> > > > 3. When you get the chance, please change the caption of the y-axis in Figure 2 to *certified accuracy* to be consistent with the terminology from Cohen et al.

---

> > > > > ### Author Response · Authors · 2022-08-06
> > > > > **Response + updated paper + thanks**
> > > > >
> > > > > First of all, we extend our sincere thanks for your continued willingness to engage with our work across the review period - it has been greatly appreciated.
> > > > >
> > > > > To respond to your points raised:
> > > > >
> > > > > 1: The boundary of $B_P(x_n, r_n)$ is $\mathcal{S}$n, but on line 109 we did also define that $r^{(n)'} = \min_{s} || S_n(s) - x_1||$. This line implies that $\mathcal{S}$ is actually the set of boundaries, which is incorrect. We also reinforced that impression with the ambiguous statement "the exterior boundary of which we describe as" which could have referred to either $B_P$ or the union of $B_P$'s.
> > > > >
> > > > > To clarify the nature of the spheres, their boundaries, and to remove ambiguity within our definitions we have added a formal definitions of $B_P$ and $\mathcal{S}$ within Defintion 3.1 (Lines 88-94); and have reframed the introduction and conclusion of Section 3.2  (Lines 109-115 and 138-145) to remove ambiguity.
> > > > >
> > > > > In aide of those updates, and using Figure 1 as a reference, $\mathcal{S}$1 is the exterior boundary of the green circle, $\mathcal{S}$2 is the exterior boundary of the red circle, and $\mathcal{S}$3 is the exterior boundary of the black circle. To provide an example about why the intersection between these surfaces takes the form of a $(d-1)$-dimensional manifolds, consider two overlapping 3D spheres. Their region of intersection is a curved manifold in 3D, but the exterior boundary of the region will be a circle, which can be thought of as being $(d-1)$-dimensional manifold. It is apparent that we were not clear enough about the $(d-1)$-dimensional manifold referred specifically to the exterior boundary of the region of intersection.
> > > > >
> > > > >
> > > > > 2: Would you be able to clarify which lines you're referring to here, as we went over lines 176-186 over the last few submitted versions of the paper and couldn't find any references to $\mathcal{S}$2 and $\mathcal{S}$3. However, it is true that the $\mathcal{S}$2 *is the surface* of $B_P(x_2, r_2)$, and we hope that our clarifications in support of Point 1) have also made this clearer.
> > > > >
> > > > > 3: Figure 2 has been updated, with thanks. We have also updated Figures 3, 6, and 7 to improve clarity regarding the median / average point raised in your earlier comment.
> > > > >
> > > > > On behalf of all the authors - thanks again, and we hope that we've reached the point of fully ameliorating your concerns (although we of course would be more than happy to answer any additional questions).

---

> > > > > > ### Comment · Reviewer_JQxG · 2022-08-06
> > > > > > **Response**
> > > > > >
> > > > > > Thank you for the latest changes! These have clarified and resolved my concerns. Also, thank you for being willing to engage and for being open to questions and feedback during this process. It takes two to tango!

---

### Meta-Review · Area_Chair_Bded · 2022-08-27

**Recommendation:** Accept
**Confidence:** Certain

**Metareview:**

The paper proposes a very simple idea that can improve one of the strongest robustness certificates. Most of the concerns were minor, and they were well addressed during the rebuttal phase. The reviewers were mostly happy with the current paper though shared a few remaining concerns, which could significantly improve the manuscript if properly addressed in the camera ready version. For instance, it will be great to see whether or not this idea can also improve the robustness certificates produced by different algorithms. The current performance gain still looks marginal, but it might be interesting if the same technique can bring in larger gains for different types of certificates.

**Award:**

No

---

### Decision · Program_Chairs · 2022-09-14

Accept